# A Building-Integrated Bifacial and Transparent PV Generator Operated by an "Under-Glass" Single Axis Solar Tracker

**Rosario Carbone *** and **Cosimo Borrello**

Dipartimento dell'Informazione, delle Infrastrutture e dell'Energia Sostenibile—"D.I.I.E.S.", University "Mediterranea" of Reggio Calabria, 89124 Reggio Calabria, Italy; brrcsm95l30h224e@studenti.unirc.it

*  Correspondence: rosario.carbone@unirc.it; Tel.: +39-0965-169-3310

**Abstract:** Nearly Zero Energy Buildings (NZEBs) play a key role in the world energy transition. This is motivating the scientific community to develop innovative electrical and thermal systems characterized by very high efficiency to specifically address the energy needs of modern buildings. Naturally, the integration of the latest generation photovoltaic (PV) systems into buildings helps to satisfy this need, and, with this objective in mind, an innovative and highly efficient building-integrated photovoltaic (BIPV) system is presented and discussed in this paper. The proposed PV system is purpose-built to be fully integrated into a variety of buildings (preferably into their rooftops) and assumes the form of a PV skylight. It is based on a certain number of innovative rotating bifacial PV modules, which are specifically made to be installed "under-glass" within a custom-made transparent casing. Thanks to their properties, the PV modules can be rotated using a very low-power, reliable, and efficient mono-axial solar tracking system, fully protected against adverse atmospheric agents. Once the proposed PV skylight is fully integrated into a building, it generates electricity and, additionally, helps to improve both the energy performance and the aesthetic appearance of the building. The electricity generation and illuminance performances of the proposed PV skylight are experimentally tested using a low-power homemade prototype driven by different solar tracking logics and under different operating conditions; the most relevant results are summarized and extensively discussed. The main outcome of the experimental study is that the most effective performance of the PV skylight is obtained by installing, in its available surface, the maximum possible number of rotating bifacial PV modules, side by side and with no empty spaces between them.

**Keywords:** NZEB; transparent BIPV; bifacial PV cells; solar trackers; PV skylights

## 1. Introduction

In order to limit the rise in global temperature to 1.5 °C, a number of countries in the world have already pledged to reach net zero emissions by 2050. To this end, the International Agency of Energy (IEA) recently published the report "Net Zero by 2050: A roadmap for the global energy system" [1]. The report sets out clear milestones for what needs to happen and when to transform the global economy from one dominated by fossil fuels into one powered predominantly by renewable energy, like solar and wind. In the building sector, the report estimates that the floor area worldwide will increase by 75% by 2050, and the electricity demand for appliances and heating/cooling equipment will continue to grow. This is because electrification, together with energy efficiency, is considered one of the main drivers of decarbonization of the building sector. In fact, despite the growth in electricity demand, total $CO_2$ emissions from the building sector are expected to decline from almost 3 Gt in 2020 to around 120 Mt in 2050, thanks to the pledged net zero pathway. According to studies and confirmed in the recent specialized literature [2,3], to achieve this, more than 85% of buildings worldwide need to comply with the zero-carbon-ready energy code by 2050 and—whenever possible—they should integrate locally

available renewable resources, e.g., geothermal, solar thermal, and solar PV. With specific reference to the PV generation of electricity, the contribution of building-integrated PV generators (BIPVs) is expected to grow from 320 TWh in 2020 to 7500 TWh in 2050.

BIPVs are simultaneously integral/structural components of building envelopes and generators of electricity (from sunlight); these solar systems are, thus, "multifunctional construction materials" [4,5]. One of the most relevant issues of BIPVs is, of course, related to the major costs of building PV solar cladding construction compared to conventional (non-PV solar) cladding construction. Nevertheless, the cost of a BIPV installation is very difficult to assess, as it is strongly influenced by a variety of factors such as the part of building skin under consideration (roof or façade), the local market characteristics, the complexity of legal and administrative procedures, etc., which are strongly variable moving from one country to another around the world. Furthermore, when the BIPV design and its installation are carried out simultaneously with other building interventions, it is possible to optimize the economic feasibility of the whole installation by significantly reducing the extra costs of BIPVs. For all these reasons, the costs of BIPVs vary widely. In particular, among the 25 real examples of BIPVs constructed worldwide and described in [6], the costs of a BIPV can range from 255 €/m$^2$ (for an in-roof installation in The Netherlands) up to 2500 €/m$^2$ (for the artistic façade of the Harbourfront Centre Theatre in Toronto). In the aforementioned report, by differentiating between roofs and façades and by excluding the most expensive examples due to their singularity, it is estimated that for a BIPV roof, the costs range from 250 €/m$^2$ to 660 €/m$^2$, while for a BIPV façade, the costs range from 280 €/m$^2$ to 850 €/m$^2$. These prices consider not only the cost of the façade/roof BIPV components but also the cost of sub-structures and materials needed to guarantee that the building is energy-efficient and the installation costs. Finally, the report also examines what happens when BIPV cladding is installed instead of regular cladding by considering both the energetic and economic benefits of BIPVs, and it summarizes that, while the costs per square meter are much higher for a BIPV cladding, after 20 years, "the total cost difference between the two solutions can almost cover the cost of a new BIPV cladding system". Of course, assessing the real price of a BIPV component remains very difficult, but the cost reduction target for BIPVs is certainly one of the keys to success in enlarging their diffusion. On the other hand, in [7], to enable a wider diffusion of innovative solutions of BIPV for both roof and façade applications, the authors highly recommend achieving high levels of ease-of-installation of BIPV elements. Furthermore, in [8], the authors present a summary of the future development of an aesthetically appealing building-integrated photovoltaic system. In [9], the authors investigate the case of a real-time adaptive BIPV shading system and its ability, in comparison with traditional static building-integrated photovoltaic façade systems, to maximize electricity generation and simultaneously ensure optimal visual comfort. Finally, in [4], the authors underline that an additional fundamental step for achieving the same goal is to maximize the energy efficiency of BIPVs within the building's energy demand because the more energy efficient the BIPV systems are, the greater their economic benefits during their lifetime. From this last point of view, in [10–13], the authors also state that innovative semi-transparent BIPV solutions can offer relevant advantages compared to conventional opaque PV modules.

With the aforementioned considerations in mind, it is easy to understand that the attractiveness (and success) of a BIPV system substantially depends on relevant factors such as its initial cost, its electrical efficiency, its thermal and lighting performances, and its aesthetic appearance. Unfortunately, compared to land-installed PV generators or building-applied photovoltaic generators (BAPVs), BIPV systems generally have a significantly higher installation cost and significantly lower electrical generation efficiency. This is because they need more sophisticated constructive materials and are forced to be installed inefficiently in terms of their exposure to sunlight. Even though BIPVs located on building façades are becoming more and more popular thanks to their visibility from the aesthetic point of view, some studies [14–16] state that the most preferred/convenient installation location for a BIPV is the rooftop of a building. This is due to both the lower installa-

tion costs and higher energy efficiency of roof-located BIPVs compared to façade-located BIPVs. In fact, due to their better solar exposure and the lower presence of shadowing phenomena, compared to façade-located BIPVs, roof-located BIPVs are characterized by good energy efficiency, especially in terms of electricity generation. Nevertheless, the roofs of buildings clearly offer smaller surface availability for the installation of BIPVs, and, as a consequence, the energy efficiency improvement of the roof-installed BIPVs, per square meter of installation surface, plays a role of primary relevance.

Solar tracking systems are potentially able to improve the electricity generation efficiency of a PV generator by up to +50% compared to the same PV generator installed in a fixed manner [17,18]. In particular, dual-axis solar tracking systems are a sophisticated and very efficient solution; nevertheless, they are expensive, complex, and could be out of order often during their lifetime due to their complexity and their exposure to aggressive atmospheric agents (wind, hail, rain, humidity). On the other hand, single-axis solar tracking systems are simpler, less sophisticated, and cheaper, and they show a greater degree of reliability and availability. However, single-axis solar tracking systems have a lower electricity generation efficiency. That value, however, can go up to about +30% compared to an equivalent fixed PV generator. Considering all the aforementioned respective characteristics, today, single-axis solar tracking systems appear to be the most widespread; nevertheless, their utilization is again limited due to expensive maintenance problems and frequent breakdowns, substantially caused by the action of adverse atmospheric agents. Recently, in the specialist literature, relevant attention has been devoted to wind load interference (and its harmful effects) on single-axis solar trackers by considering both single-row solar trackers and solar tracker arrays [19–23]. The main outcomes of all these theoretical and experimental studies confirm that single-axis solar trackers are prone, by themselves, to very harmful wind-induced interferences and damaging torsional vibrations.

Bifacial photovoltaic cell and module technologies are rapidly increasing their market shares, and it now seems possible that, in the future, much of the bifacial PV cell production will be monofacial modules realized with white back encapsulants and/or reflective backsheets to enhance their front side power rating [24,25]. Among the practical applications of bifacial PV cell technology, one of the most hopeful is pairing it with the use of single-axis solar tracking systems [26,27]. In [26], the authors presented preliminary research concerning floating tracking photovoltaic systems; the study evaluates the energy performances of different PV plant configurations located in mid- and high-latitudes. From the main outcomes of the study, the authors concluded that, with respect to conventional (ground and fixed) PV plants, the gain due to tracking, natural cooling, and bifaciality for two mid- and high-latitude localities is, respectively, 31.3% and 27.8% for the single-axis vertical systems and 47.4% and 42.5% for the dual-axis systems. However, the authors deferred to future work for the economic analysis of the different plant configurations. Similarly, in [27], the authors presented a comparative analysis of the yield potential and cost-effectiveness of different kinds of PV plants installed worldwide. In particular, they used data validated from real, worldwide PV installations and from the literature to perform a comparative analysis of installation and maintenance costs and performances between fixed-tilt PV plants and PV plants based on single- and dual-axis tracking systems; furthermore, they considered both monofacial and bifacial PV cell technologies. The results of the comparative analysis revealed that bifacial PV cell technology paired with a single-axis solar tracking installation can increase energy yield by 35% and that this solution reaches the lowest LCOE for the majority of locations (93.1% of the land area). On the contrary, although dual-axis solar tracking installations achieved the highest energy generation, their costs and complexity are still too high and are, therefore, not as cost-effective.

Regardless of their category, when speaking about solar tracking systems, it is important to underline an additional and very relevant issue. That is to say, if, on the one hand, solar tracking systems promise a relevant improvement in PV generation efficiency (in comparison with an equivalent fixed PV generator), on the other hand, they need—in practice—to occupy a greater available surface for installing the same PV modules of its

equivalent fixed PV generator (i.e., in the case of a multi-string PV generator). In fact, when a rotating PV generator is made using a certain number of PV strings (which have to be rotated in unison to track the sun), in order to avoid the reciprocal shadowing phenomena during their daily rotations, the PV strings have to be sufficiently distanced from each other. Therefore, a PV generator that uses a single-axis solar tracking system needs to occupy a greater land surface compared with that needed for a fixed installation with the same number of PV modules. Obviously, this corresponds to installing less PV power per square meter of available land surface. Obviously, in the sector of BIPVs, this last aspect has great relevance because, in this specific case, the available installation surfaces well-exposed to sunlight are a very precious and rare resource.

Starting from a basic idea introduced and explained by the authors in an Italian patent dated 2018 [28] and also considering some new ideas proposed by the authors in a pending Italian patent application dated 2023 [29], this paper introduces an innovative transparent BIPV system, operated by using a special single-axis solar tracking system which is (i) specifically made to be easily and effectively integrated into the roofs of a variety of buildings and (ii) fully protected from the adverse atmospheric agents "under-glass". First, this BIPV system is constructed from a number of custom-designed bifacial PV modules in the form of a single row of series-connected bifacial PV cells. Then, the bifacial PV modules are installed side by side in a special transparent casing, which is specifically made to be a structural part of the rooftop of the building. This transparent casing is designed to contain inside all the electro-mechanical constitutive elements of the single-axis solar tracking system, completely eliminating the main issues deriving from the aggressive actions of atmospheric agents. In this way, the proposed transparent BIPV system assumes the form of an innovative PV skylight. Depending on the specific energy and aesthetic needs of the building, the entire surface of the rooftop of a building can be fully utilized for the installation of more than one of the proposed PV skylights, which could also be constructed in a modular form.

In the following sections, some theoretical and practical aspects that can affect the optimal design criteria of the proposed PV skylight are introduced and discussed. Then, the constitutive characteristics of the proposed PV skylight are detailed and discussed together with its energy, illuminance, and aesthetic potential performances. Finally, a low-power homemade prototype, specifically designed and constructed for experimental and scientific investigation purposes, is introduced and described, and the results of a series of experimental tests are exposed and deeply discussed. At the end of the paper, the authors summarize the most relevant experimental results, aiming to give insight into the most realistic and effective form for the future definitive and commercial version of the introduced PV skylight; also, some considerations about possible and future studies are developed.

## 2. Methods and Materials

The subject of this study is a PV skylight specifically designed to be fully integrated on the rooftop of buildings to improve their overall energetic performances together with their aesthetic appearance. To achieve the aforementioned goals, the PV skylight should possess, at the same time, different features. Among them, three features are very relevant: (i) it should be able to illuminate and/or shade the underlying surface/volume of the building in which it is integrated; (ii) it should be able to control the illuminance and/or the shadowing of the underlying surface/volume of the building in which it is integrated; (iii) it should be highly efficient in generating electricity.

As already mentioned in the introduction, the proposed PV skylight incorporates a special single-axis solar tracker, which is the basis for all the aforementioned features.

Deferring to the next sections for a detailed description of the constitutive characteristics of the proposed PV skylight, in this section, some relevant issues concerning the optimal installation of the rotating strings of a generic PV plant equipped with a single-axis solar tracker are briefly addressed, from both the theoretical and the practical points of view. This should help to optimally design and implement the proposed skylight.

When installing a certain number of PV strings on a conventional land single-axis solar tracking system, it is a widespread practice to look for the installation configuration that maximizes the electricity generation of each rotating PV string. To obtain this reasonable goal, it is necessary to avoid any reciprocal shadowing phenomenon that can occur during the daily rotations of the PV strings, which are necessarily installed side by side [30]. Furthermore, it is well known that avoiding reciprocal shadowing phenomena among the PV strings helps to avoid any adverse hotspot phenomenon and maintain their lifespan [31]. Nevertheless, the only way to avoid (or limit at the minimum) the aforementioned shadowing phenomena is to space the rotating PV strings properly; that is to say, an empty space must be introduced between each pair of PV strings installed side by side. In this way, the installation of all the rotating PV strings occupies more land with respect to the minimum strictly needed. From a different point of view, this also means that the PV power installed per square meter of the available land is significantly reduced and, as a consequence, the electricity generation per square meter of the available land could be significantly reduced compared to different possible installation configurations (i.e., the installation of all the PV strings in a fixed manner and occupying the entire available land area). Obviously, the final result depends, on the one hand, on the gain in electricity generation obtained for each installed, rotating, and unshaded PV string and, on the other hand, on the loss of PV power installed per square meter of land caused by the need to avoid the shadowing.

With the increasing number of PV installations worldwide, surface areas that are well-exposed to sunlight are becoming more and more precious, especially in the sector of BIPVs. From this point of view, it could be relevant to deeply investigate the most effective solution for obtaining the maximum electricity generation per square meter of the available surface area when installing a mono-axial solar tracking system of an array of PV strings installed side by side.

Deferring to a future study for a more general, detailed, theoretical and experimental analysis, in the following, for the sake of brevity, only some first considerations are developed, with the specific aim of understanding the most effective installation configuration for the realization of the PV skylight under study.

In [30], the aforementioned problem has been addressed in a very similar way; nevertheless, in the paper, no specific attention is devoted to the maximization of the electricity generated by the PV system per square meter of the occupied terrestrial land, as is the specific interest of this study. The paper demonstrates that, in principle, the aforementioned problem can also be addressed with an analytical approach; in fact, in its case, the authors introduced an interesting "spatial projection method".

Confining the problem to only the maximization of the electricity generation of each PV string, which composes a greater PV array to be installed in a certain available surface area, some first details about the analytical approach can be provided for understanding how complex the more general problem could be. Starting from the availability of a certain land surface, whose width is named $W_t$, and from the availability of a certain number of rotating PV strings to be installed, whose width is named $W_i$, the main objective of the analysis is to determine the installation configuration (position and number of the PV strings) that guarantees the maximum generation of electricity. Also, starting from a preliminary analysis of the geographic location and orographic characteristics of the installation site, the maximum effective rotation angle of the rotating PV strings can be estimated, named $\alpha_m$. As stated in [30], the most effective installation configuration is the one that guarantees the maximum electricity generation of each rotating PV string, that is to say, the one assuring that no reciprocal shadowing phenomena occur among the rotating PV strings installed side by side. This installation can be implemented by simply introducing an empty space, named $E_s$, between each pair of rotating PV strings, whose minimum values have to be properly estimated starting from the abovementioned fixed data. Avoiding more detail, in the case of a plane land area of installation, the minimum value of $E_s$ can be simply calculated by the following formula:

$$E_s = W_i * (1 - \cos \cdot \alpha_m)/\cos \cdot \alpha_m \tag{1}$$

Once calculated, $E_s$, the maximum number, $n_{max}$, of the rotating PV strings that can be installed on the available surface area (having a total width equal to $W_t$), can also be easily calculated. Finally, the electricity generated by the $n_{max}$ rotating and unshaded PV strings can be theoretically estimated.

On the other hand, even if the reciprocal shadowing phenomena among the rotating PV strings reduce the daily electricity generation of each (and could also cause adverse hotspot phenomena), it is not possible to exclude that by installing more rotating PV strings side by side and with a reduced empty space between each pair of them, the new PV installation could generate more total electricity (the sum of the electricity of the higher number of the installed rotating shaded PV strings), compared to the configuration installation based on a reduced number of unshaded rotating PV strings. In principle, this additional issue could be completed and preliminarily investigated using a theoretical approach; nevertheless, the analytic formulation (and solution) results are very complex and based on some unrealistic exemplary hypotheses. Avoiding more details, first, it has to be considered that the maximum number of rotating PV strings that can accept a certain level of reciprocal shading is a function of the percentage reduction of $E_s$, with respect to its minimum value that guarantees the absence of shadowing. Furthermore, the reduction in the electricity generation of each shadowed rotating PV string is proportional to the percentage value of its shaded surface area, and this last is a function of the percentage reduction in $E_s$ and time. Finally, a theoretical formula could be derived under unrealistic exemplary hypotheses (e.g., no reflected or diffused sunlight affects the rotating PV modules).

Considering the complexity and the high degree of approximation of the aforementioned theoretical approach, as an alternative, the relevant issue of looking for the maximum generation of electricity per square meter of an available land surface can probably be more effectively studied and tested experimentally; that is to say, by field measurements on properly implemented test case studies.

As stated above, in order to design the most effective configuration of the PV skylight under study, in addition to its electricity generation performance, the illuminance and/or shadowing capacity and the capacity to effectively control these latter performance parameters must also be considered. From this point of view, it is easy to understand that the higher the number of rotating PV modules installed on the available surface, the more the capacity of the PV skylight to shadow the underlying surface/volume of the building in which it is installed, also with easy and effective controllability. On the contrary, the lower the number of rotating PV modules installed on the available surface, the higher the capacity of the PV skylight to illuminate the underlying surface/volume of the building in which it is installed. Even if it might appear that the choice of the number of rotating PV modules to install on the PV skylight surface is a controversial issue, please consider that, when needed, a high level of illuminance of the underlying surface/volume can be obtained by installing a high number of rotating PV modules, by simply implementing a specific rotation control logic of the PV modules (as in conventional window shutters) [29].

Considering the aforementioned theoretical and practical aspects, after discussing the basic features of the PV generators introduced in [28,29], to understand the optimal design and installation configuration of the PV skylight under study, in the following, an experimental series of measurements is presented and discussed, rather than a theoretical formulation of the problem. The experimental study is fully based on the utilization of a special homemade prototype, specifically designed and constructed to make possible the implementation of different operating conditions and installation configurations.

## 3. Constitutive Characteristics of the Proposed PV Skylight

The basic idea for obtaining a transparent PV generator that can be fully integrated into a variety of buildings, exploiting the well-known benefits of a single-axis solar tracker, has already been introduced by the authors in an Italian patent [28]. Starting with these contents and from the considerations developed in the previous section, this section describes the executive design and the basic constitutive characteristics of an innovative PV

skylight, specifically conceived to be fully integrated onto the rooftop of a building. Unlike conventional PV skylights, which are its direct competitors, the proposed one is equipped with a special single-axis solar tracker fully protected "under-glass". As already underlined, it is based on the utilization of a certain number of special single-row rotating bifacial PV modules characterized by a very low weight and thickness.

### 3.1. The Rotating Bifacial, Single-Row, Low-Thickness PV Modules

In order to obtain, at the same time, a transparent PV generator fully integrable into a variety of buildings and equipped with a single-axis solar tracker, first of all, as basic components, a certain number of custom-designed rotating bifacial single-row PV modules are introduced. These PV modules are constructed in the form of a single row of a certain number of series-connected bifacial PV cells; once constructed, they are installed side by side within a custom-designed transparent casing, which is designed to be fully integrated into a building (preferably on its rooftop). All the electro-mechanical components of the special single-axis solar tracking systems are also installed within the same transparent casing so they are fully protected against all atmospheric agents.

To conduct an experimental and scientific investigation about the potential contribution of the bifaciality of the rotating PV modules, each bifacial PV module of the prototype is constructed using two separate single-row PV strings of monofacial PV cells that are properly encapsulated, shoulder to shoulder. In this way, the currents generated by the two faces of the bifacial PV module (the front face and the rear face) can be independently measured and analyzed. In other words, for experimental and scientific study purposes, when constructing the prototypes, the use of commercial bifacial PV cells was avoided, and instead, commercial monofacial PV cells were used and assembled in the aforementioned "shoulder to shoulder" arrangement. In practice, to construct each bifacial PV module, two PV strings, each based on three series-connected PV cells, were used, therefore utilizing, de facto, six monofacial PV cells. Once constructed, the two separate and identical PV strings were glued shoulder to shoulder with each other to obtain the desired bifacial PV module, with its front face and its rear face physically separated and characterized by the same generation capacity. Thanks to this, during our experimental tests, for each bifacial rotating PV module, we could separately analyze the attitude of its front face and its rear face in generating electricity under certain (and variable) sunlight exposure conditions.

Figure 1 graphically summarizes the construction process of the homemade prototype of a bifacial single-row PV module.

Figure 1a refers to the first step, where two identical and separate single-row PV strings are constructed using three series-connected monofacial PV cells before they are glued shoulder to shoulder. Figure 1b refers to the second step, where the two PV strings are encapsulated, shoulder to shoulder, between two transparent plastic thin films (a front sheet and a back sheet) using an EVA material as glue in the middle. Figure 1c is a photo of the bifacial single-row PV module prototype before the mounting of its special terminations, while Figure 1d is a photo that shows the PV module in its final form.

Using the construction procedure described above, the obtained single-row bifacial PV module is simultaneously light, slim, semi-rigid, and ready to rotate (around its fictitious central axis) when mounted within its transparent casing; additionally, the currents generated by its front face and by its rear face can be measured, monitored, and analyzed independently.

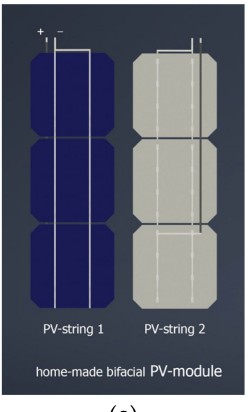
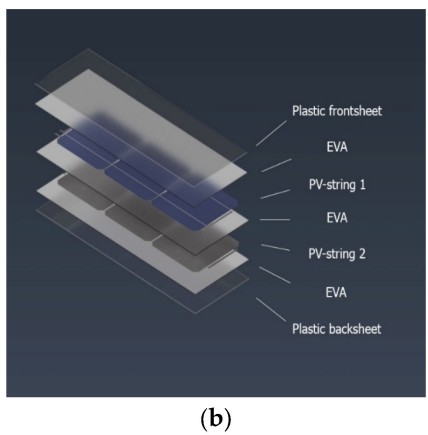
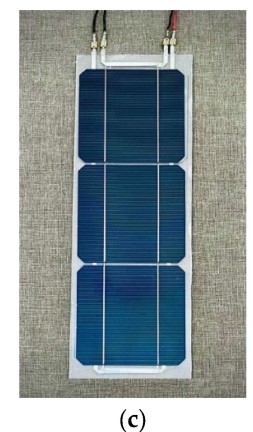
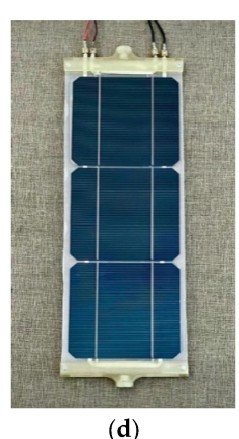

(**a**) (**b**) (**c**) (**d**)

**Figure 1.** Realization of a prototype of a single-row bifacial PV module, based on two separate and identical monofacial PV strings, glued shoulder to shoulder with each other; each PV string is constructed using three series-connected monofacial PV cells: (**a**) the two PV strings, before they are glued shoulder to shoulder with each other; (**b**) stratigraphy of the bifacial PV module; (**c**) a real photo of the bifacial PV module prototype before the mounting of its special terminations; (**d**) a real photo of the bifacial PV module prototype in its final form.

Please note that, outside the specific objectives of this academic experimental study, for the construction of the above-described bifacial single-row PV modules, for eventual commercial purposes, we obviously suggest the use of single bifacial PV strings built based on the most efficient commercial bifacial PV cells.

### 3.2. Installing the Single-Row Bifacial PV Module within Its Transparent Casing

Once a certain number of single-row bifacial PV modules have been built, they must be installed inside their custom-designed transparent protective casing. As can be easily deduced from Figure 1d, each single-row PV module appears able to rotate around its fictitious central longitudinal axis, which is identified by the imaginary line that connects the central pins of its special terminations. This characteristic makes building a reliable and cheap mono-axial solar tracking system inside the same casing very simple, which we are now discussing. Basically, the geometry and the constitutive materials of the transparent casing must be designed and selected so that it can be easily and fully integrated into the structure of a certain building (i.e., its rooftop). Furthermore, its perimeter shoulders and its transparent cover must be designed in order to guarantee the installation of all the single-row bifacial PV modules in the form of side-by-side rows properly spaced from each other so that two important factors are always guaranteed during the daily rotations of the PV modules. These two factors are (i) the generation of the maximum available electrical power and (ii) a certain degree of transparency in the PV skylight to the incident sunlight and its easy controllability. From this point of view, the optimal solution should be an empty space between two rotating PV modules installed side by side, and this empty space should be large enough to avoid any reciprocal shading among the PV modules during their daily rotations, even for a large rotation angle. In fact, in this way, it should be possible to maximize both the electricity generation capacity and the degree of transparency to the incident sunlight of the PV skylight. Once defined, the approximative sizing of the transparent casing, considering the total number of the rotating PV modules to be installed inside of it and the width of the aforementioned empty interspaces, its geometry has to be executively defined considering the characteristics (materials and sizes) of all the components of the single-axis tracking system. Thanks to the reduced number, dimension, and weight of the rotating PV modules, a single low-power (just a few watts) stepper motor should be enough to rotate (in unison) all the PV modules of a "medium-size" PV skylight. Furthermore, as detailed in the section dedicated to the description of the prototypes utilized for the experimental tests, the electronic components (microcontroller,

motor drivers, sensors) are few and occupy a low volume. As a consequence, the rotating PV modules, the motor, the electronics, and the additional components of the mechanical transmission system of the solar tracker can be installed within a small dedicated "service box" located just below one of the two sides of the casing, perpendicular to the rotation axes of the PV modules. Figure 2 shows the final aesthetic appearance of an example (with three or five rotating PV modules) of the proposed PV skylight, and it should also better explain the constitutive additional characteristics of its transparent casing.

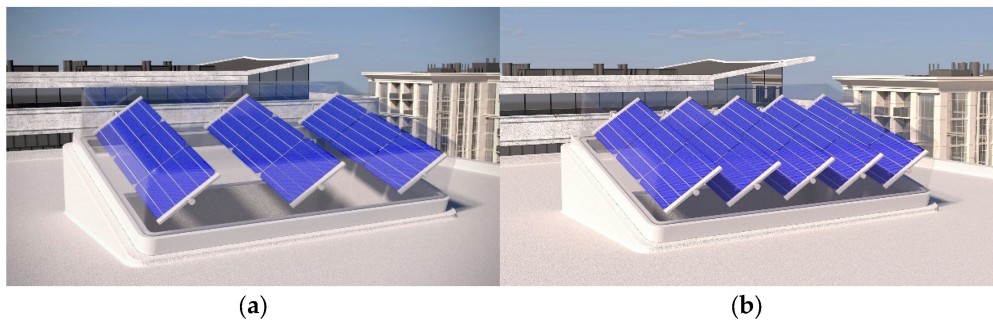

(**a**)                                                                                 (**b**)

**Figure 2.** A photo-realistic representation of an example of the proposed PV skylight, based on three (**a**) or five (**b**) single-row rotating bifacial PV modules installed side by side.

Finally, Figure 3 gives an idea of the feasibility and the practical utilization of the proposed PV skylight by showing that it can be integrated on the rooftop of a building, improving both its energy and illuminance performance together with its aesthetic appearance.

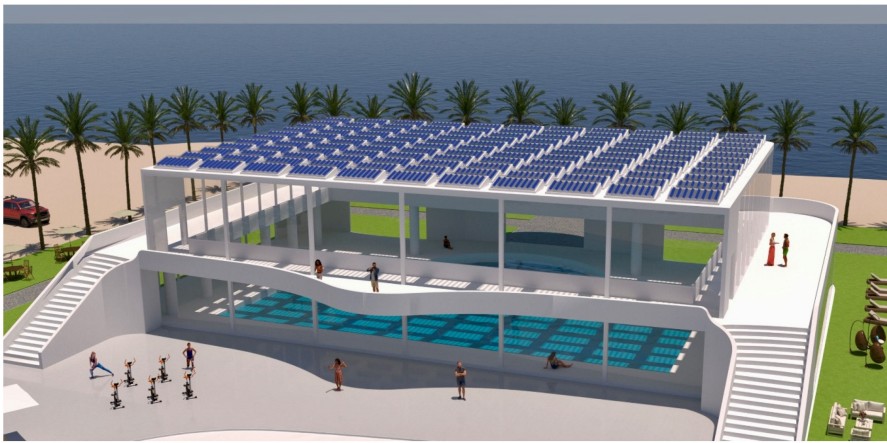

**Figure 3.** A photo-realistic representation of an example of integration of the proposed PV skylight on the rooftop of a building.

## 4. The Homemade Prototype and the Experimental Investigation on the Potential Performances of the Proposed PV Skylight

### 4.1. Premise

Before introducing a detailed description of the prototype designed and constructed for developing the experimental tests, some basic theoretical considerations are briefly developed about the criteria for designing the PV skylight and for investigating its potential performance.

Obviously, the main objective of installing a PV skylight in a building is the improvement of both its energetic and illuminance performance, together with its aesthetic appearance.

Starting from this assumption, one of the most important performance parameters of a PV skylight is certainly the maximum generation of electricity, according to the exposure of the building to sunlight. Therefore, the transparency of the PV skylight and its controllability are very important.

Considering that the proposed PV skylight also includes a single-axis solar tracking system, in principle (as underlined in Section 2), the more intuitive solution for obtaining

the best performance of the PV skylight by installing only a limited number of rotating PV modules, guaranteeing enough empty space between them on the available surface exposed to the sunlight. In this way, it is possible to guarantee both the maximum electrical power generation (by avoiding mutual shading effects among rotating PV modules) and the maximum transparency of the PV skylight. On the other hand, this solution could have some shortcomings. First, limiting the number of rotating bifacial PV modules mounted on the available surface area of the PV skylight limits the total PV power installed on the available surface. Furthermore, if the rotating PV modules are obliged to always track (in unison) the sun's position perpendicularly (to maximize the electrical power), no control is dedicated to the transparency of the PV skylight. Finally, from the academic point of view, limiting the number of the rotating PV modules mounted on the test prototype would not make it possible to fully investigate the potential performance of the PV skylights under study and/or to compare them with conventional PV skylights. Furthermore, as already underlined in Section 2, it is difficult to theoretically predict the most effective installation configuration of the rotating PV modules to obtain a PV skylight that is efficient and, at the same time, characterized by high functional flexibility. For all these reasons, the homemade prototype has been specifically designed to have a high degree of flexibility so that it can be used to experimentally emulate the behavior of the proposed PV skylight under different exposures to the sunlight for different numbers of rotating PV modules installed on its available surface and for different sun tracking control logics. More details on the constructive and functional characteristics of the prototype and on the performed experimental investigation are specified and discussed in the Sections 4.2–4.4

### 4.2. Constitutive Characteristics of the PV Generator Prototype

Figures 1 and 2 illustrate the constitutive components and the construction steps for building our proposed PV skylight. In particular, Figure 2 shows the structural characteristics of its transparent casing in order to fully integrate the PV skylight into the rooftop of a building. Figure 2a also advises the readers about the "reasonable" prospect of not fully covering the entire available surface area of the PV skylight in order to preserve a significant degree of its transparency to the incident sunlight and to avoid mutual shading effects on its rotating PV modules. Figure 2b also shows that fully covering the available surface of the PV skylight with more rotating PV modules could be more beneficial overall. Therefore, in order to identify the optimal configuration of the PV skylight and the optimal control logic of its rotating bifacial PV modules, the prototype used for performing the experimental tests was specifically designed and constructed with some special features; this is illustrated in Figure 4, with the help of two photos.

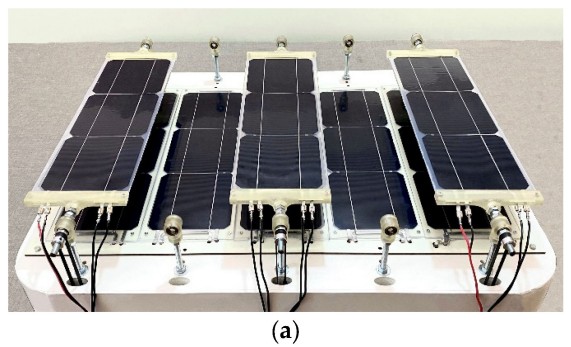
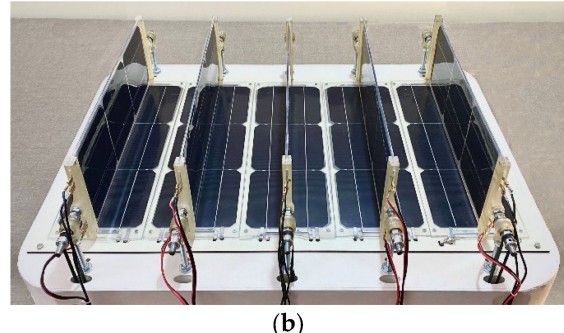

(**a**)                                                                                           (**b**)

**Figure 4.** Two different photos of the PV generator prototype used for the experimental tests, with three (**a**) or five (**b**) rotating bifacial PV modules.

With the help of Figure 4, it is possible to underline the following main constitutive characteristics of the test prototype:

- The prototype is based on a total of five identical single-row rotating bifacial PV modules;

- Each rotating bifacial PV module has two identical (and separate) faces (the front and the rear); each one is realized by connecting in series three (5 × 5)″ mono-crystalline PV cells, and these two faces can be electrically monitored separately;
- The five rotating bifacial PV modules can be installed side by side with no empty space between them, and the prototype's total surface area exposed to sunlight is equal to five times that of a single PV module;
- For test purposes, each of the five PV modules is equipped with its respective low-power stepper motor; they can be rotated independently from each other, can rotate from 0° to 180°, and, if desired, can rotate in unison with all the remaining PV modules;
- Each rotating bifacial PV module can be easily removed from the prototype in order to make any desired changes to their number and their reciprocal position on the PV skylight.

Some additional characteristics and specifications of its main components are reported in Table 1.

**Table 1.** Technical specifications of the main components of the prototype.

| Component | Description |
|---|---|
| PV cells | $I_{SC}$ = 6.03 A, $V_{OC}$ = 0.64 V, $P_{mpp\ (STC)}$ = 3.07 $W_p$, $\eta_{max}$ = 19.2% |
| Terminations | 3D custom printed, with clear resin |
| Motors | Nema mod. 17HS08-1004S |
| Transmission | Based on GT2 belts and pulleys |

*4.3. Characteristics of the Data Acquisition System and the Electronic Control System*

In fixed-environment conditions (intensity of the sunlight, ambient temperature, cloudiness, ventilation, and so on), the power that can be generated by a PV module is directly proportional to its short-circuit current; the stronger its short-circuit current, the higher its generable electrical power. Obviously, the electrical power actually generated by the PV module also depends on its electrical load condition. Nevertheless, if one assumes that the load condition is always made optimal by an ongoing and uninterruptible operation of a specific maximum power point tracker (MPPT)—as is normally the case—the measurement of the short-circuit current of a PV module corresponds to an indirect measurement of the maximum electrical power that the PV module can generate.

Starting from the aforementioned assumption, first, the two PV strings of each rotating PV module (mounted on its front face and on its rear face) are permanently short-circuited, and their respective short-circuit currents are constantly measured, acquired, and stored. This means that, for measuring all the short-circuit currents, our prototype counts ten amperemeters (two for each of the five rotating PV modules). These short-circuit currents are constantly acquired and stored, and they are also made accessible on the web for any numerical and/or graphical post-processing needs.

Concerning the tracking of the solar position each day and the consequent control of the daily rotation of the PV modules, it is relevant that the main objective of the experimental investigation is to understand how the position of the rotating PV modules affects the performance of the proposed PV skylight not only in terms of the maximum electrical power it can generate but also in terms of incident sunlight that can pass through it (i.e., in terms of its transparency) and also in terms of the controllability of this last performance parameter.

For this reason, a "special" control logic for the rotation of the five PV modules during an entire day has also been implemented. Before giving more details about this control logic, it is relevant to restate that the solar tracker of the proposed PV skylight has only one axis and, as a consequence, the prototype is installed with a fixed tilt angle and with a fixed north–south terrestrial direction; also, the five step motors of the solar tracker are controlled so that the corresponding five rotating PV modules of the prototype rotate in unison, from east to west.

The aforementioned special control logic is based on a certain number of repetitive "rotation sequences", which are imposed on all five rotating PV modules during an entire day. The first rotation sequence starts at sunrise, while the last rotation sequence ends at sunset. In practice, over a whole day, the first rotation sequence is periodically repeated about every five minutes until the end of the day.

Each rotation sequence consists of four different steps, which are specified in detail below.

In step (i), the five PV modules are placed with their surfaces parallel to the ground, with an angular position of 90°, like a conventional fixed PV skylight (with no solar tracker), as in Figure 4a; the five PV modules stay in this position for about two minutes.

In the next step (ii), all five PV modules in unison start a "quick" counterclockwise rotation that ends when their surfaces are perpendicular to the ground, with an angular position of 0°, as in Figure 4b.

In the next step (iii), all five PV modules start a new, "slow" clockwise rotation (from 0° to 180°), which ends when the surfaces of the five PV modules are perpendicular to the ground again.

In the next step (iv), all five PV modules "quickly" come back to the initial position parallel to the ground, and they stay there for about two minutes before repeating steps (ii), (iii), and (iv).

During the aforementioned daily rotation sequences, all the short-circuit currents (two for each of the five rotating PV modules) are constantly measured, acquired, and stored to be numerically post-processed off-line; in this way, one can read, analyze, and understand how the position of the PV modules over an entire day, affects (i) the electricity generation of its front face and its rear face (and also their sum), (ii) the transparency of the PV skylight, and (iii) the controllability of the transparency of the PV skylight.

Some additional characteristics and specifications about the main components of the embedded system utilized for the motor driving, for the measurements, and for the data acquisition and upload are reported in Table 2.

**Table 2.** Technical specifications of the main components of the homemade embedded system.

| Components | Model |
|---|---|
| Microcontroller | ESP32-WROOM 32 |
| Amperemeters | ACS712 20 A |
| ADCs | Ads1115 16 bit |
| Motor drivers | TMC2209 v1.2 |

*4.4. Description of the Homemade Illuminance Sensor, Created for Estimating the Transparency Degree of the PV Skylight*

In the previous sections, two additional and important performance parameters of the PV skylight have been cited and underlined: its transparency to the incident sunlight and the degree of controllability of this transparency. For experimentally investigating the variation in the aforementioned performance parameters during the daily rotation sequences of the five rotating PV modules described above, a useful homemade illuminance sensor was conceived and constructed to estimate the level of the incident sunlight that, passing through the PV skylight, arrives on the underlying surface. This illuminance sensor is based on five additional fixed (and monofacial) PV modules, identical to each other. Each of these additional PV modules has a geometry and a surface area identical to each of the rotating bifacial PV modules. Furthermore, these additional PV modules are constructed from the same (5 × 5)″ mono-crystalline PV cells already utilized for the construction of the rotating bifacial PV modules. First, the five additional monofacial PV modules are glued on a low-thickness rigid plastic foil, side by side and with no empty spaces between each other; furthermore, they are electrically independent. In this way, they create a monofacial plane PV surface identical to the front surface created by the five rotating bifacial PV modules

when they are placed at the angular position of 90° (parallel to the ground). Finally, as shown in Figure 4, this resulting flat PV surface can be installed exactly under the surface area of the PV skylight (also at different heights, if necessary for study purposes). Figure 5 specifically illustrates the aforementioned homemade flat PV surface with a photograph.

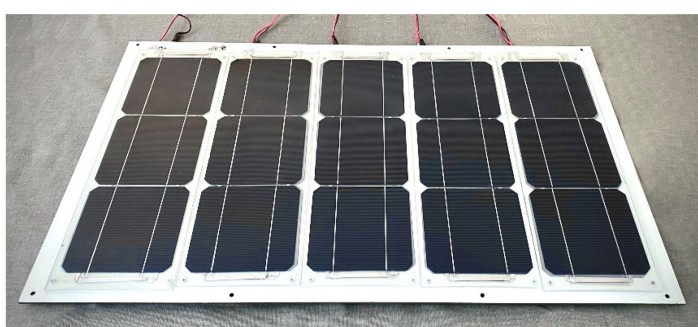

**Figure 5.** A photo of the homemade illuminance sensor (PV surface).

Similar to the five bifacial and rotating PV modules, the five additional monofacial PV modules are also permanently short-circuited, and its additional five short-circuit currents are permanently measured, acquired, and stored using five additional amperemeters and the previously cited data acquisition system.

In practice, to estimate the intensity of the sunlight that passes through the PV skylight and its uniformity, we measure the five different short-circuit currents generated by the five fixed monofacial PV modules of our PV surface, which is installed exactly under the PV skylight surface area, assuming that the sunlight that strikes the five PV modules of the underlying PV surface is proportional to their respective measured short-circuit currents.

## 5. Experimental Results and Discussion

For emulating and testing the behavior of the proposed PV skylight and its potential performance, the homemade prototype described in the previous section was used to perform a series of experimental measurements, both in terms of electricity generation and controllability of its transparency degree to the incident sunlight. The experiments were performed under different operating conditions. In particular, considering that the prototype can mount a maximum of five rotating bifacial PV modules on its available surface area exposed to sunlight, the prototype was tested by implementing two different installation configurations. In the first case study, only three rotating bifacial PV modules were installed on the prototype, visibly interspaced one from each other (as shown in Figure 4a); in the second case study, all five rotating bifacial PV modules were installed on the prototype, side by side with no empty spaces between them (as shown in Figure 4b). Please consider that all the following experimental tests were performed on the terrace of a building located in the city of Reggio Calabria, in the south of Italy. The aforementioned case studies are discussed in detail in the following two separate sub-sections.

### 5.1. First Case Study: Testing the Prototype Based on Three Rotating Bifacial PV Modules Interspaced One from Each Other

The photograph in Figure 4a illustrates the operating conditions of the prototype (emulating a PV skylight) when it is ready to be utilized for performing the first set of experimental tests.

In practice, on the entire surface area of the prototype (equal to five times the surface area of a single PV module), we installed only three rotating bifacial PV modules. In particular, between the first lateral (M1) and the second central (M2) rotating PV module and between the second (M2) and the third lateral (M3) rotating PV module, an empty space is left, equal to the width of each rotating PV module; that is to say, only the 3/5 of the entire surface area of the PV skylight was utilized for installing the three rotating PV modules while 2/5 of it was left empty. This is to guarantee a very limited time of

reciprocal shadowing phenomena between the three rotating PV modules during their daily rotations and, as a consequence, to guarantee, as much as possible, improvement in the electricity generation of each of them, together with a high degree of transparency of the PV skylight to the incident sunlight.

As already specified in Section 4.3, thanks to the single-axis solar tracker and its electronic control system, all three rotating PV modules (M1, M2, and M3) rotate in unison by implementing the already-specified four-step repetitive rotation sequence, conceived ad hoc.

At the first step of the aforementioned rotation sequence, the PV modules are parallel to the ground (as the PV modules of a conventional semitransparent fixed PV skylight), and they remain in this 90° angular position for about 2 min.

In the second step, all the PV modules rapidly rotate counterclockwise (in just a few seconds) to reach the 0° angle position, where they are perpendicular to the ground.

In the third step, all the PV modules slowly rotate clockwise from the 0° angle position to the 180° angle position until they are perpendicular to the ground again. This step takes about two minutes. During the rotation, all eleven short-circuit currents of the eight PV modules installed on the prototype are constantly measured, acquired, and stored by the homemade data acquisition system. Please note that the prototype mounts three rotating bifacial PV modules. This means that six (2 × 3) short-circuit currents have to be measured; additionally, the homemade illuminance sensor (the PV surface shown in Figure 5) consists of five additional monofacial PV modules (M4, M5, M6, M7, and M8) and this means that five additional short-circuit currents have to be measured, for a total of eleven short-circuit currents.

In the fourth step, the PV modules rapidly rotate counterclockwise (in just a few seconds) to reach the initial 90° angle position. They remain in this position for about two minutes before restarting the same four-step rotation sequence again.

In practice, over an entire day, each rotation sequence is repeated every 5 min, from sunrise to sunset.

First, in order to appreciate the practical and scientific usefulness of the implemented rotation sequences, Figure 6 shows an excerpt of the entire waveforms of some of the measured and registered short-circuit currents. In particular, the excerpt in Figure 6 specifically refers to the central (M2) rotating bifacial PV module of the prototype and also of the central fixed monofacial PV module (M6) of the underlying PV surface (that is, de facto, the illuminance sensor); the data were measured and registered during the cloudless day of 21 July 2023.

In more detail, for all four steps of the already-specified rotation sequence, Figure 6 reports the waveforms of (a) the short-circuit current of the front face of the central M2 rotating bifacial PV module, $I_{sh-F}$; (b) the short-circuit current of the rear face of the M2 rotating bifacial PV module, $I_{sh-R}$; (c) the sum of the two aforementioned currents, $I_{sh-S}$; (d) the short-circuit current of the fixed monofacial central PV module M6 of the underlying PV surface, $I_{sh-U}$. As specified also in the time axis of the figure, this excerpt refers to a single rotation sequence operated in the early morning of the test day.

From the analysis of the waveforms reported in Figure 6, it is possible to develop some interesting considerations. During step (i) of the rotation sequence, when the central rotating bifacial PV module of the prototype, M2, is parallel to the ground (like the fixed PV modules of a conventional semitransparent PV skylight), its front face generates about 3.1 amps while its rear face generates a very reduced current of about 0.5 amps. This means that the contribution of the bifaciality of the M2 PV module, in this position and at this time, results to be about +16%. In the same step, thanks to the "visibly" empty space between the PV module M1 and the PV module M2, in the central zone of the underlying PV surface, the illuminance level is relatively high (the respective central PV module M6 generates about 2.3 amps, that is to say almost 75% of the short-circuit current generated by the front face of the rotating PV module M2). At the end of the quick step (ii), the rotating bifacial PV module M2 reaches the position perpendicular to the ground, and the current generated by its front face increases from 3.1 amps to 4 amps, while the current generated by its rear face remains practically constant. At the same time, the illuminance on the underlying

PV module M6 decreases slightly (the respective current decreases from 2.3 amps to 2.1 amps). During the slow step (iii), the current generated by the front face of the rotating PV module M2 continues to increase, and it reaches its maximum value of about 4.9 amps at the angular position of about 40°. Please note that because the angular speed of the motors is constant, the rotation angle varies (from 0° to 180°) linearly with time. Also, in the early morning, the position at which the current generated by the front face of the rotating PV module M2 reaches its maximum value is not perpendicular to the position of the sun rays because, at this time, the first lateral PV module M1 projects a shadow onto PV module M2 (under analysis). On the other hand, when PV module M2 reaches the angular position (40°) at which the aforementioned shadow disappears, it generates an actual current lower than the theoretical maximum value that it could have generated in the absence of PV module M1 at the angular position perpendicular to the sun rays of 33° (for instance, the front face of the lateral PV module M1, at the angular position perpendicular to the sun rays of 33°, generates a maximum short-circuit current of about 5.3 amps). Regarding the contribution of the bifaciality of the rotating PV module M2 to its whole generation capacity, please note that the maximum value of $I_{sh-S}$ (the sum of the two short-circuit currents $I_{sh-F}$ and $I_{sh-R}$) does not occur at the same angular position of the maximum of $I_{sh-F}$, and it is equal to about 5.4 amps, that is to say, +10% with respect to the maximum value of $I_{sh-F}$. Furthermore, the illuminance level on the underlying PV surface ($I_{sh-U}$) results in a minimum value of about 1.45 amps, in correspondence with the maximum value of $I_{sh-F}$, while it assumes a higher value of about 1.55 amps, in correspondence with the maximum value of $I_{sh-S}$. Finally, please note that the illuminance level of the underlying PV module M6, $I_{sh-U}$, reaches its maximum value of about 3.4 amps in correspondence with the angular position of the rotating PV modules at about 150°; in this angular position, the rotating PV modules are practically "parallel" to the incident sun rays and the whole electricity generation of M2 ($I_{sh-S}$) is equal to the noteworthy value of about 2.5 amps (46% of its maximum value). During step (iv), the rotating PV modules quickly return to the initial 90° angle position, and they remain there for about two minutes before starting a new rotation sequence.

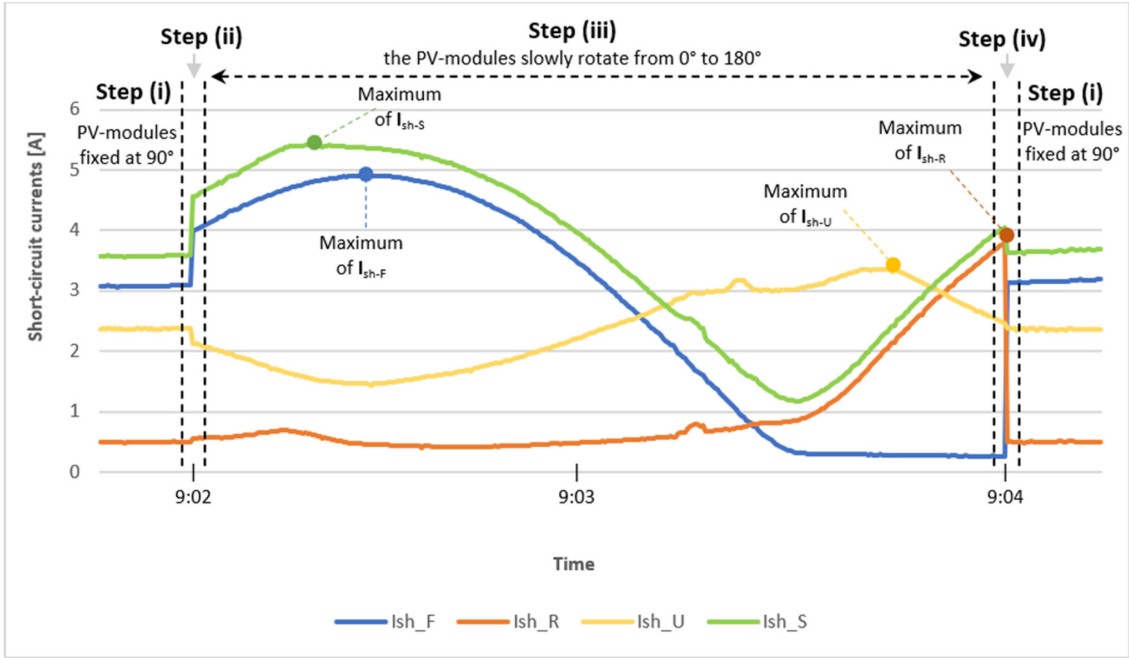

**Figure 6.** Waveforms of the short-circuit currents of the central rotating PV module M2 ($I_{sh-F}$, $I_{sh-R}$, $I_{sh-S}$) and of the central fixed PV module M6 of the underlying PV surface ($I_{sh-U}$), during the duration time of a single and complete (four-step) rotation sequence, implemented in the early morning of the cloudless day of 21 July 2023, as extracted from the respective whole daily waveforms.

To give a more complete representation of what happens over the whole day, Figure 7 reports four additional excerpts from the daily waveforms of the same aforementioned short-circuit currents; these additional excerpts refer (a) to the late morning, (b) to midday, (c) to the early afternoon, and (d) to the late afternoon. Even if the waveforms are discernably different from those of Figure 6, it is easy to extend the detailed analysis already developed for the previous waveforms independently to them.

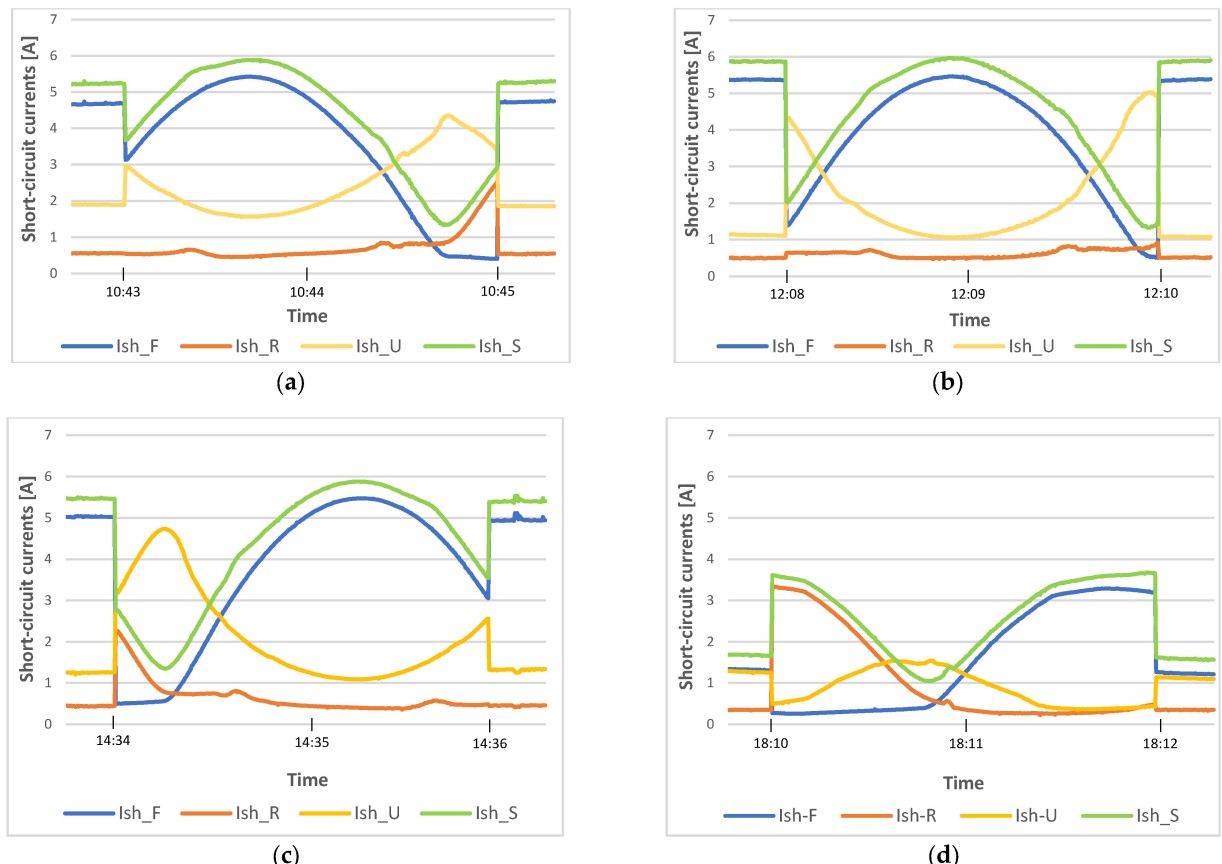

**Figure 7.** Waveforms of the short-circuit currents $I_{sh-F}$, $I_{sh-R}$, $I_{sh-S}$, and $I_{sh-U}$ previously defined in Figure 6, during the duration time of a single and complete rotation sequence: (**a**) in the late morning; (**b**) at midday; (**c**) in the early afternoon; (**d**) in the late afternoon.

Once the daily whole waveforms of all the short-circuit currents (of all the rotating bifacial PV modules and all the fixed monofacial PV modules of the underlying PV surface) were measured, acquired, and registered and after analyzing their contents during each four-step rotation sequence, numerical post-processing was performed in order to derive some additional important information about the potential performance of the PV skylight under study. The main results of this last analysis are reported in Figure 8.

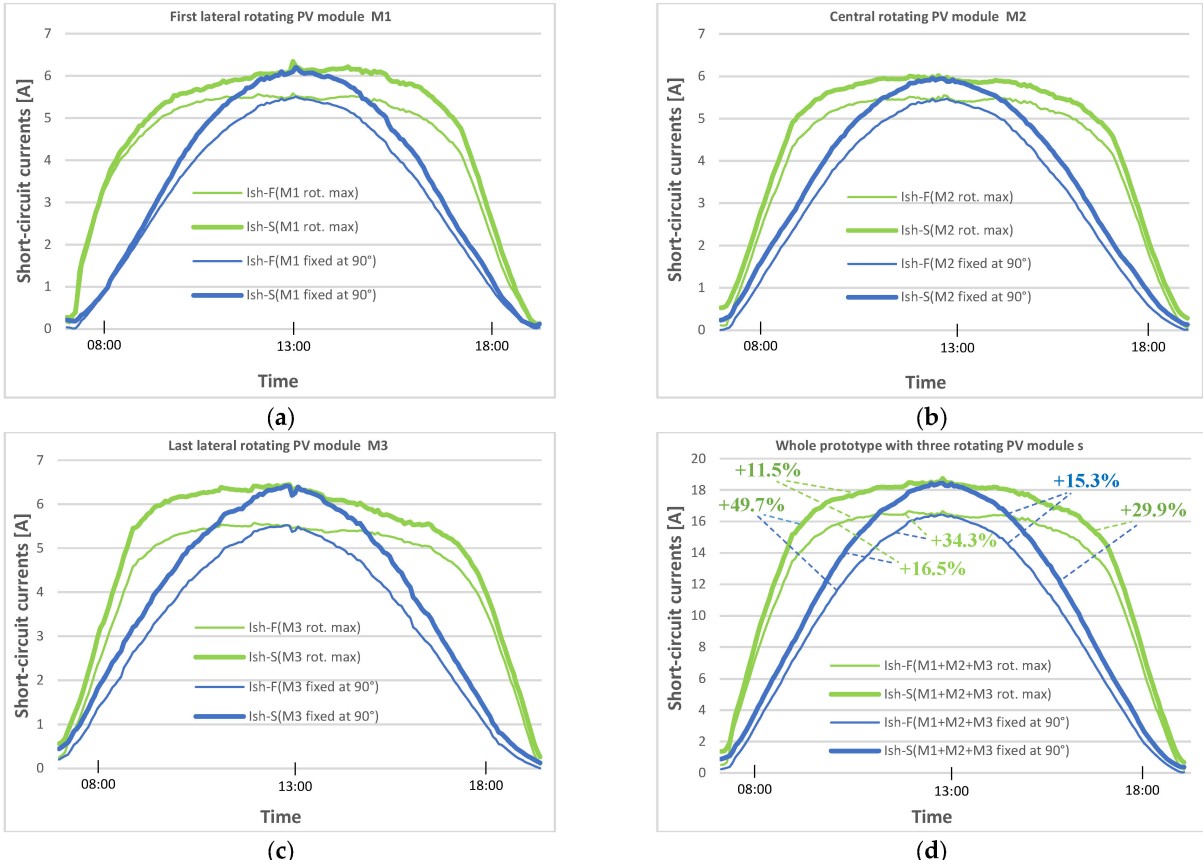

**Figure 8.** Daily curves of the short-circuit currents of the three rotating PV modules M1, M2, and M3, from the picture (**a**) to the picture (**c**), and of the whole prototype, in the picture (**d**). Each picture refers to (i) the front face of the *i*-th PV module when it is fixed at the 90° angle position, $I_{sh-F(Mi\ fixed\ at\ 90°)}$; (ii) the sum of the front face and of the rear face of the *i*-th PV module, when it is fixed at the 90° angle position, $I_{sh-S(Mi\ fixed\ at\ 90°)}$; (iii) the maximum value of the front face of the *i*-th PV module, measured during each rotation sequence, $I_{sh-F(Mi\ max)}$; (iv) the maximum values of the sum of the front face and of the rear face of the *i*-th PV module, measured during each rotation sequence, $I_{sh-S(Mi\ max)}$.

The most relevant outcomes can be deduced from Figure 8d. It emerges that (i) for a fixed semitransparent bifacial PV skylight, the bifaciality of the PV modules guarantees an improvement in the generable electricity of +15.3%, compared to the same fixed monofacial PV skylight; (ii) for the rotating semitransparent bifacial PV skylight, the bifaciality of the PV modules guarantees an improvement in the generable electricity of +11.5%, compared to the same rotating monofacial PV skylight; (iii) the rotating semitransparent monofacial PV skylight can generate +34.3% more electrical power compared to a fixed semitransparent monofacial PV skylight and +16.5% compared to a fixed semitransparent bifacial PV skylight; (iv) the rotating semitransparent bifacial PV skylight can generate +49.7% more electrical power than a fixed semitransparent monofacial PV skylight and +29.9% compared to a fixed semitransparent bifacial PV skylight.

Please note that even if the highest performing rotating bifacial PV skylight promises a great improvement (+29.9%) in electrical power generation compared to a fixed semitransparent bifacial PV skylight, it does not reach the value of the electrical energy that could be generated by installing, on the same available surface area of the PV skylight, five fixed monofacial PV modules, that is to say by installing an almost "opaque" fixed PV skylight. In fact, considering just the medium generation capacity of each fixed monofacial PV module of the tested prototype, it is easy to estimate that the electrical power generable by the aforementioned opaque fixed monofacial PV skylight constructed with five fixed PV

modules could be more than +11% higher than that generable by the highest performing rotating semitransparent bifacial PV skylight.

Considering this relevant aspect, it remains evident that the global advantage of the rotating semitransparent bifacial PV skylight compared to a fixed "opaque" monofacial PV skylight is its degree of transparency to the incident sunlight and its controllability. This last point is analyzed in Figure 9.

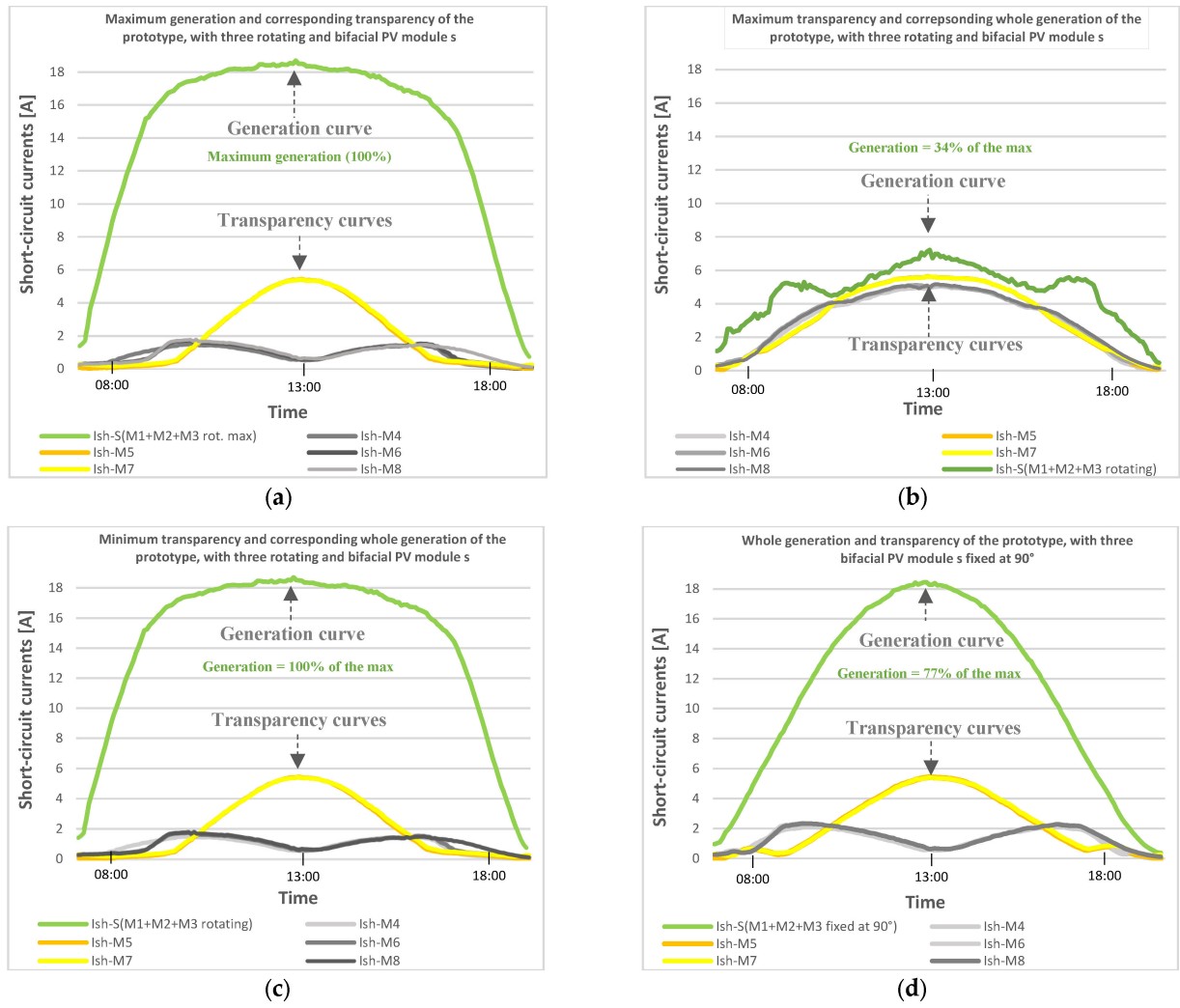

**Figure 9.** Daily curves of the five short-circuit currents, $I_{sh-M4}$, $I_{sh-M5}$, $I_{sh-M6}$, $I_{sh-M7}$, and $I_{sh-M8}$, of the five fixed monofacial PV modules of the underlying PV surface (M4, M5, M6, M7, and M8) and of the three rotating bifacial PV modules, $I_{sh-S(M1 + M2 + M3 \text{ rot. max})}$, under four different operating conditions. The picture (**a**) refers to the operating condition in which the three rotating PV-modules are rotated for catching the maximum electrical power, during the entire test day. The picture (**b**) refers to the operating condition in which the three rotating PV-modules remained fixed at the 90° angular position for the entire test day. The picture (**c**) refers to the operating condition in which the three rotating PV-modules are rotated for catching the maximum transparency of the prototype, during the entire test day. The picture (**d**) refers to the operating condition in which the three rotating PV-modules are rotated for catching the minimum transparency of the prototype, during the entire test day.

Figure 9 analyzes the waveforms of five short-circuit currents of the corresponding five fixed monofacial PV modules of the underlying PV surface, utilized as illuminance sensors; these currents are also related to the daily whole current generated by the PV skylight. From these waveforms, it is possible to determine the illuminance degree and its

variation in the five respective zones of the underlying surface during an entire day. Four different operating conditions of practical relevance are considered.

Figure 9a plots the waveforms of the five short-circuit currents of the five PV modules of the underlying PV surface (M4, M5, M6, M7, and M8) when all the three rotating bifacial PV modules (M1, M2, and M3) are rotated to catch the maximum electrical power.

Figure 9b plots the waveforms of the five short-circuit currents of the five PV modules of the underlying PV surface (M4, M5, M6, M7, and M8) when all the three rotating bifacial PV modules (M1, M2, and M3) are rotated to obtain the maximum transparency degree of the PV skylight.

Figure 9c plots the waveforms of the five short-circuit currents of the five PV modules of the underlying PV surface (M4, M5, M6, M7, and M8) when all the three rotating bifacial PV modules (M1, M2, and M3) are rotated to obtain the minimum transparency degree of the PV skylight (that is to say, the maximum shadowing of the underlying surface).

In order to make a comparative analysis with a semitransparent fixed PV skylight, Figure 9d also plots the waveforms of the five short-circuit currents of the same five PV modules of the underlying PV surface (M4, M5, M6, M7, and M8) when all the three rotating bifacial PV modules (M1, M2 and M3) are maintained fixed at the 90° angle position.

The main outcomes of the analysis of Figure 9, from (a) to (d), can be summarized as follows.

The prototype controlled specifically for obtaining the maximum generation of electricity (Figure 9a) shows that the corresponding PV skylight also has a good medium degree of transparency to the incident sunlight. Nevertheless, the corresponding illuminance level on the underlying surface is not uniform; in fact, the three zones exactly under the three rotating PV modules are less illuminated with respect to the two zones below the two empty spaces of the overlying surface, which are very illuminated.

The prototype specifically controlled for obtaining the maximum degree of transparency (Figure 9b) shows that it is possible to make the prototype "fully transparent" (like a conventional glass skylight with no PV generation) by specifically controlling its rotating PV modules. In fact, the curves of the short-circuit currents of the PV modules (M4 ÷ M8) of the underlying PV surface show that they can generate practically the same currents of the overlying rotating PV modules when these last are fixed at the 90° angle position. Please note that this operating condition can be easily obtained by controlling the angular position of all the rotating PV modules so that they are constantly "parallel" to the incident sun rays. However, in this operating condition, the daily electricity generation of the PV skylight decreases to about 34% of the maximum generable value.

The prototype specifically controlled to obtain the minimum degree of transparency (that is to say, the maximum shadowing of the underlying surface, Figure 9c) reveals that this specific PV skylight is not capable of becoming "opaque" to the incident sunlight. In fact, to obtain this performance, the only possibility is controlling the rotating PV modules so that they are constantly "perpendicular" to the incident sun rays. Because of the presence of the two empty spaces between the three rotating PV modules, this is not sufficient to prevent incident sunlight from passing through the PV skylight. Furthermore, please note that this operating condition coincides with that of the maximum generation of electricity.

Finally, the prototype that emulates a fixed semitransparent PV skylight (Figure 9d) shows that this generates only 77% of the maximum generable electricity. Furthermore, its degree of transparency to the incident sunlight and the uniformity of the corresponding illuminance level on the underlying surface are practically the same as those of the rotating PV skylight generating the maximum electrical power.

Starting from the aforementioned outcomes, in the next section, we introduce a new series of measurements developed for exploring the possibility of improving the performance of the proposed PV skylight by installing two additional rotating bifacial PV modules on the prototype to occupy the entire available surface exposed to sunlight.

### 5.2. Second Case Study: Testing the Prototype Based on Five Rotating Bifacial PV Modules, Installed Side by Side with No Empty Spaces

The photograph in Figure 4b illustrates the operating conditions of the prototype (emulating a PV skylight) when it is ready to be utilized for performing a new set of experimental tests. In particular, also considering the theoretical premise reported in Section 4.1, these new experimental tests are finalized to determine if the performance of the proposed PV skylight can be improved when compared with those of the previous solution based on only three rotating PV modules, both in terms of maximum electricity generation and of control of the sunlight passing through the PV skylight. In practice, on the entire surface area of the prototype (equal to five times the surface area of a single rotating PV module), we installed five rotating bifacial PV modules (M1, M2, M3, M4, and M5). Obviously, this time, the aforementioned PV modules were installed on the prototype side by side with no empty spaces between them; that is to say, all 5/5 of the available surface area of the PV skylight was occupied by the rotating PV modules.

As specified in Section 4.3, thanks to the single-axis solar tracker and its electronic control system, all five rotating PV modules rotate in unison by implementing the four-step repetitive daily rotation sequence already described in detail in previous sections. The same homemade PV surface, based on five additional fixed monofacial PV modules (now named M6, M7, M8, M9, and M10), is again utilized for analyzing the sunlight illuminance level on the surface under the PV skylight. Also, the daily rotation sequences, together with the measurement method and the data acquisition system, are the same as the previous case study.

First, in order to appreciate the practical and scientific usefulness of the implemented rotation sequences in this operating condition, Figure 10 shows an excerpt of the whole waveforms of the rotating bifacial PV module M2 of the prototype and also of the underlying fixed monofacial PV module M7; the data were measured and registered during the cloudless day of 22 July 2023 at the same site as the previous case study.

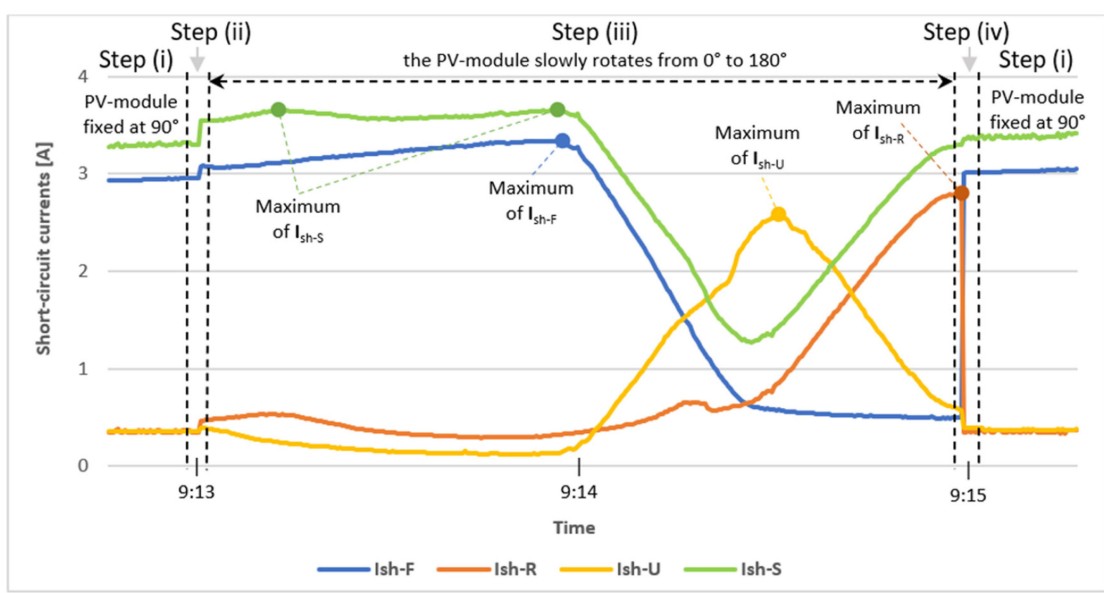

**Figure 10.** Waveforms of the short-circuit currents of the rotating PV module M2 ($I_{sh-F}$, $I_{sh-R}$, $I_{sh-S}$) and of the fixed PV module M7 of the underlying PV surface ($I_{sh-U}$) during the duration time of a single and complete (four-step) rotation sequence, implemented in the early morning of the cloudless day of 22 July 2023, as extracted from the respective whole daily waveforms.

In more detail, for all four steps of the already-specified daily rotation sequence, Figure 10 reports the waveforms of (a) the short-circuit current of the front face of the rotating bifacial PV module M2, $I_{sh-F}$; (b) the short-circuit current of the rear face of the rotating bifacial PV module M2, $I_{sh-R}$; (c) the sum of the two aforementioned currents,

$I_{sh-S}$; (d) the short-circuit current of the fixed monofacial PV module M7 of the underlying PV surface, $I_{sh-U}$. As specified in the time axis of the figure, this excerpt refers to a single rotation sequence operated in the early morning of the test day.

For this case study, from the analysis of the waveforms reported in Figure 10, it is possible to develop some first interesting considerations.

During step (i) of the rotation sequence, when the rotating bifacial PV module M2 of the prototype is parallel to the ground (like the fixed PV modules of a conventional "opaque" PV skylight), its front face generates about 2.9 amps, while its rear face generates a very reduced current of about 0.4 amps. This means that the contribution of the bifaciality to the whole generation of the M2 PV module, in this position and at this time, results to be about +14%. In the same step, considering that this time there are no empty spaces between the five rotating PV modules, on the underlying zone of the PV surface, the illuminance level is very low; in fact, its respective PV module M7 generates about 0.4 amps.

At the end of the quick step (ii), the rotating bifacial PV module M2 reaches the position perpendicular to the ground, and its front face slightly increases the generated current, from 2.9 amps to 3.1 amps, and also the current generated by its rear face increases slightly, from 0.4 amps to 0.5 amps. At the same time, the illuminance level on the underlying PV module M7 remains practically constant; in fact, its respective current remains at the low value of 0.4 amps.

During the slow step (iii), the current generated by the front face of the rotating PV module M2, $I_{sh-F}$, increases very slowly, and it reaches its maximum value of about 3.4 amps in the new angular position of about 88°. This means that the rotation of the PV module M2 contributes little to the improvement of its generation capacity because, at this time, the first lateral PV module M1 projects a strong shadow on the PV module M2 (under analysis); on the other hand, when the aforementioned shadow disappears, the actual current generated by the PV module M2 registers a relevant loss with respect to the theoretical maximum value that it could have generated in the absence of the shadowing PV module M1 in the angular position perpendicular to the sun rays (for instance, at the same step, the front face of the first lateral PV module M1 generates a maximum short-circuit current of about 4.7 amps, at the angular position perpendicular to the sun rays of about 35°). Regarding the contribution of the bifaciality of the rotating PV module M2 to its whole generation capacity, please note that the maximum value of $I_{sh-S}$ (the sum of the two short-circuit currents $I_{sh-F}$ and $I_{sh-R}$) does not occur at the same angular position of the maximum of $I_{sh-F,}$ and it is equal to about 3.7 amps, that is to say, it increases about +9% with respect to the maximum value of $I_{sh-F}$. Furthermore, in this position, the illuminance level on the underlying PV surface ($I_{sh-U}$) is at its minimum very low value of about 0.15 amps. Finally, please note that the illuminance level of the underlying PV module M7 reaches its maximum and relevant value of about 2.6 amps in correspondence with the angular position of about 137°; in this angular position, the aforementioned PV module is practically "parallel" to the incident sun rays and its entire electricity generation ($I_{sh-S}$) is equal to the value of about 1.5 amps (that is to say, 40.5% of its maximum value).

During step (iv), the rotating PV modules quickly return to the initial 90° angle position, and they remain there for about two minutes before starting a new rotation sequence.

To give a more complete representation of what happens during the whole day, Figure 11 reports four additional excerpts from the daily waveforms of the same aforementioned short-circuit currents; these additional excerpts refer (a) to the late morning, (b) to midday, (c) to the early afternoon, and (d) to the late afternoon. Even if the waveforms are discernably different from Figure 10, it is easy to extend them independently to the detailed analysis already developed for the previous waveforms.

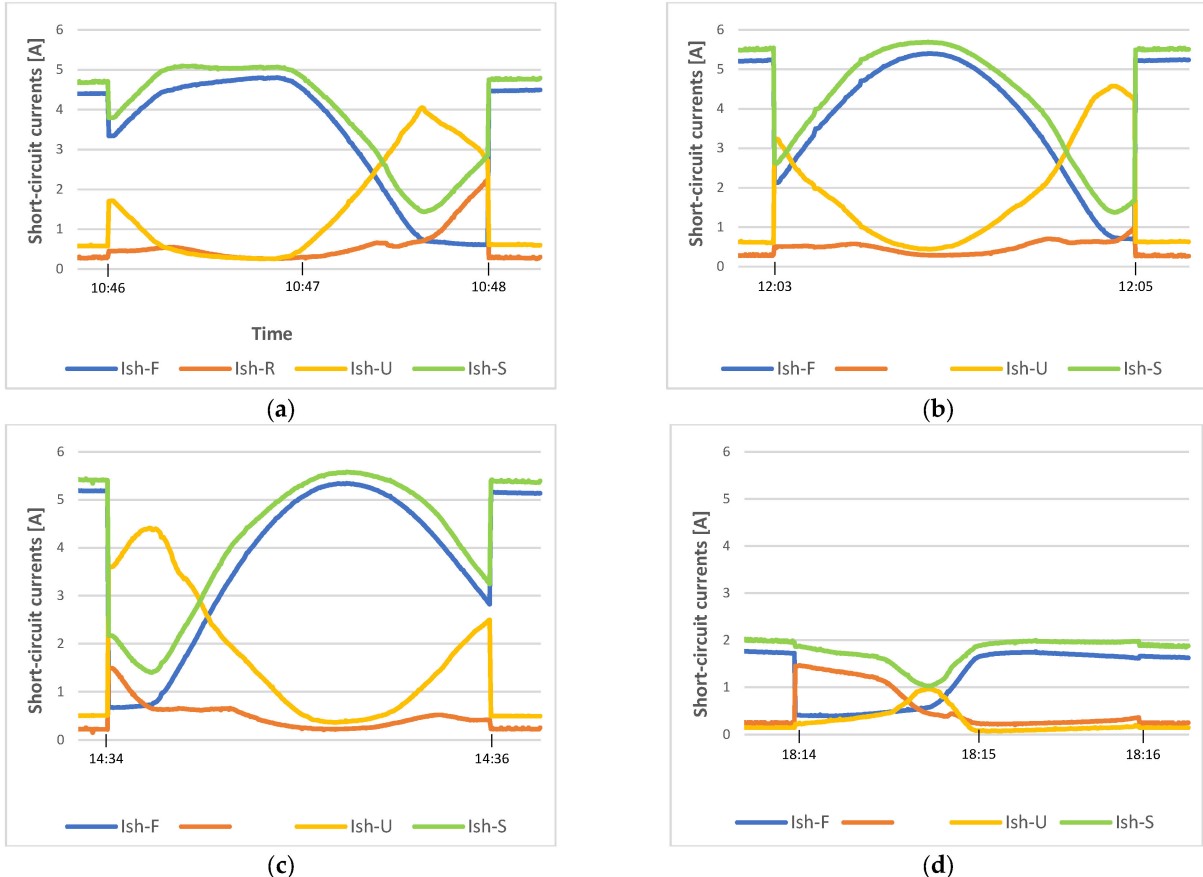

**Figure 11.** Waveforms of the short-circuit currents $I_{sh-F}$, $I_{sh-R}$, $I_{sh-S}$, and $I_{sh-U}$ previously defined in Figure 10, during the duration time of a single and complete rotation sequence, as extracted from (**a**) the late morning; (**b**) midday; (**c**) the early afternoon; (**d**) the late afternoon.

Once the daily whole waveforms of all the short-circuit currents (of all five rotating bifacial PV modules and all five fixed monofacial PV modules of the underlying PV surface) were measured, acquired, and registered and after analyzing their contents during each four-step rotation sequence of the test day, numerical post-processing performed in order to derive some additional important information about the potential performance of the introduced PV skylight. The main results of this last analysis are reported in Figure 12.

The most relevant outcomes can be seen in Figure 12f. It emerges that (i) for a fixed bifacial PV skylight, the bifaciality of its PV modules guarantees an improvement in the generable electricity of +9.6%, compared to the same fixed monofacial PV skylight; (ii) for the rotating bifacial PV skylight, the bifaciality of its PV modules guarantees an improvement in the generable electricity of +10.5%, compared to the same rotating monofacial PV skylight; (iii) the rotating monofacial PV skylight can generate +10.6% more electrical power compared to a fixed monofacial PV skylight and +0.9% compared to a fixed bifacial PV skylight; (iv) the rotating bifacial PV skylight can generate +22.1% more electrical power compared to a fixed monofacial PV skylight and +11.4% compared to a fixed bifacial PV skylight.

In more detail, the PV skylight constructed using five rotating bifacial PV modules installed side by side without any empty space from each other promises a relevant improvement in the generable electricity with respect to both the monofacial fixed five PV module ("opaque") PV skylight (+22%) and the bifacial fixed five PV module ("opaque") PV skylight (+11%). Furthermore, compared with the rotating bifacial PV skylight created using three PV modules, analyzed in the Section 5.1, the PV skylight with five rotating bifacial PV modules promises an improvement in the maximum generable electricity of +40.5%. As a consequence, it is possible to state that the PV skylight constructed with five

rotating bifacial PV modules is the most high-performing solution in terms of maximum electricity generation.

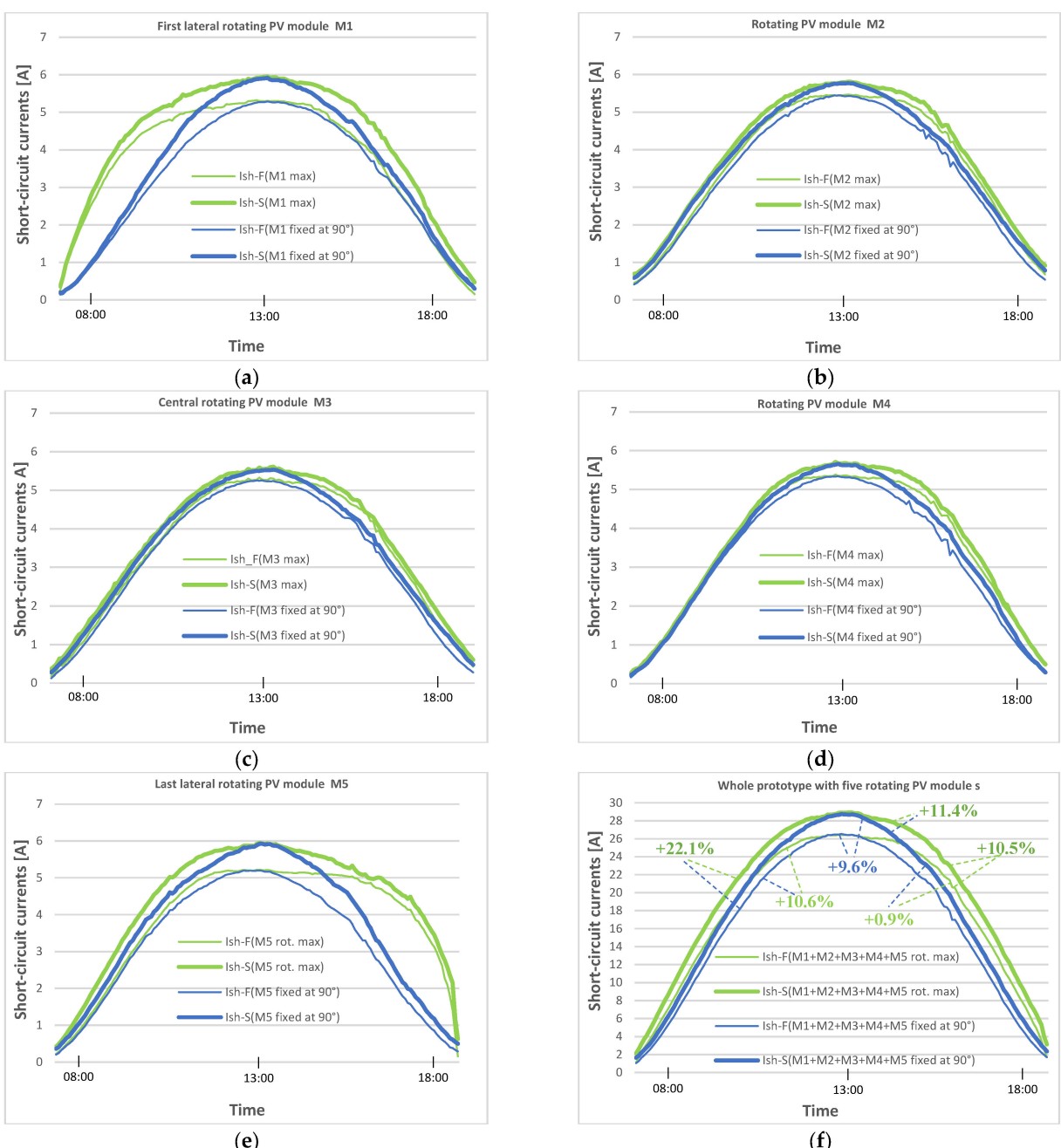

**Figure 12.** Daily curves of the short-circuit currents of the five rotating PV modules M1, M2, M3, M4, and M5, from picture (**a**) to picture (**e**) respectively, and of the whole prototype, picture (**f**). Each picture refers to (i) the front face of the *i*-th rotating bifacial PV module when it is fixed at the 90° angle position, $I_{sh-F(Mi\ fixed\ at\ 90°)}$; (ii) the sum of the front face and of the rear face of the *i*-th PV module, when it is fixed at the 90° angle position, $I_{sh-S(Mi\ fixed\ at\ 90°)}$; (iii) the maximum value of the front face of the *i*-th PV module, measured during each rotation sequence, $I_{sh-F(Mi\ rot.\ max)}$; (iv) the maximum values of the sum of the front face and of the rear face of the *i*-th PV module, measured during each rotation sequence, $I_{sh-S(Mi\ rot.\ max)}$.

To experimentally analyze the degree and controllability of the transparency of this configuration of the PV skylight under study, Figure 13 reports the waveforms of the five short-circuit currents of the corresponding five fixed monofacial PV modules of the

underlying PV surface (utilized as an illuminance sensor); these currents are also related with the daily whole current generated by the PV skylight. From these waveforms, it is possible to determine the illuminance degree and its variation on the five respective zones of the underlying surface during an entire day. Four different operating conditions of practical relevance are considered.

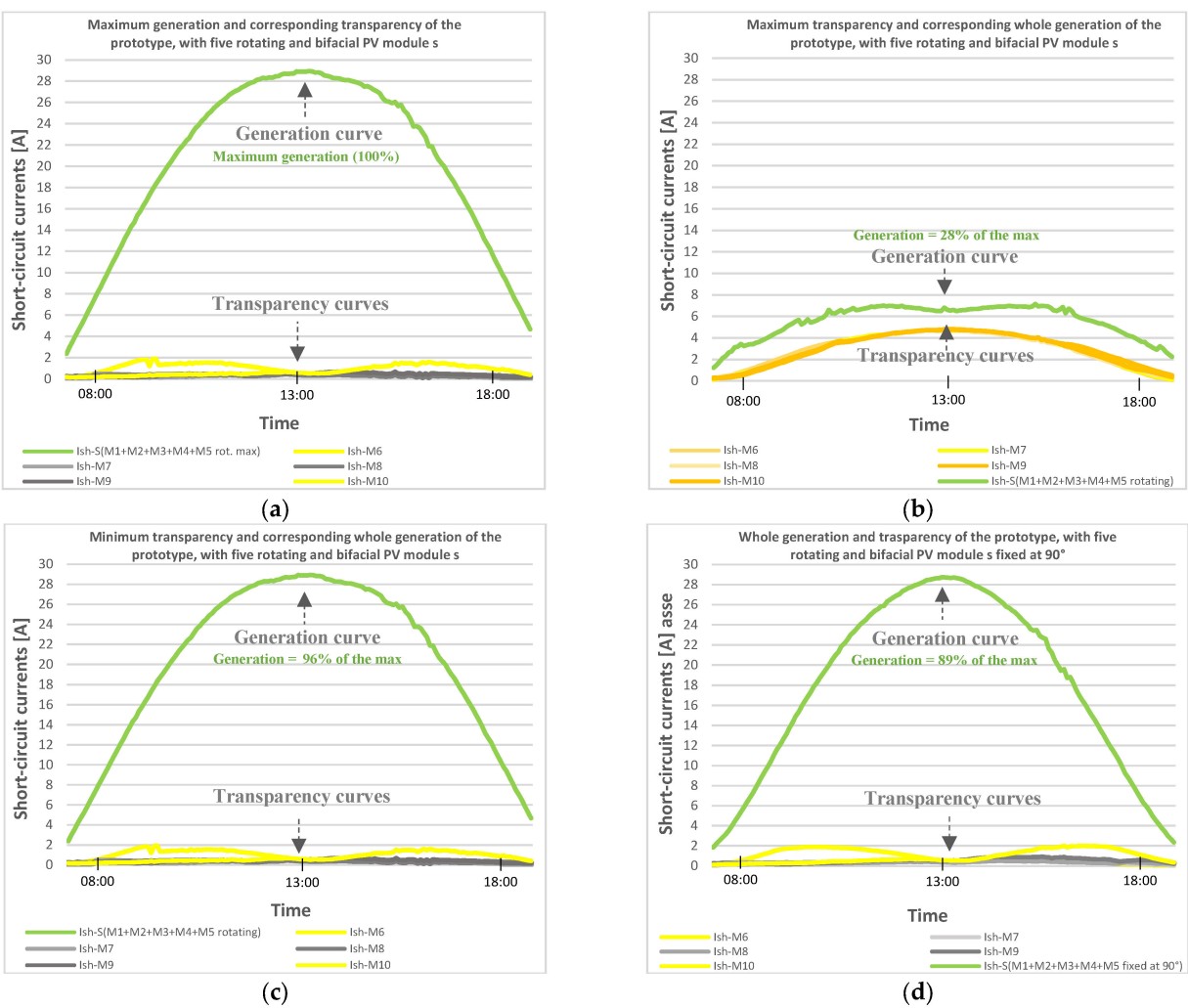

**Figure 13.** Daily curves of the five short-circuit currents, $I_{sh-M6}$, $I_{sh-M7}$, $I_{sh-M8}$, $I_{sh-M9}$, and $I_{sh-M10}$, of the five fixed monofacial PV modules of the underlying PV surface (M6, M7, M8, M9, and M10) and of the five rotating bifacial PV modules, $I_{sh-S(M1 + M2 + M3 + M4 + M5\ rot.\ max)}$, under four different operating conditions. The picture (**a**) refers to the operating condition in which the five rotating PV-modules are rotated for catching the maximum electrical power, during the entire test day. The picture (**b**) refers to the operating condition in which the five rotating PV-modules remained fixed at the 90° angular position for the entire test day. The picture (**c**) refers to the operating condition in which the five rotating PV-modules are rotated for catching the maximum transparency of the prototype, during the entire test day. The picture (**d**) refers to the operating condition in which the five rotating PV-modules are rotated for catching the minimum transparency of the prototype, during the entire test day.

Figure 13a plots the waveforms of the five short-circuit currents of the five PV modules of the underlying PV surface (now named M6, M7, M8, M9, and M10) when all five rotating bifacial PV modules (M1, M2, M3, M4, and M5) are rotated to catch the maximum electrical power.

Figure 13b plots the waveforms of the five short-circuit currents of the five PV modules of the underlying PV surface (M6, M7, M8, M9, and M10) when all five rotating bifacial

PV modules (M1, M2, M3, M4, and M5) are rotated to obtain the maximum transparency degree of the PV skylight.

Figure 13c plots the waveforms of the five short-circuit currents of the five PV modules of the underlying PV surface (M6, M7, M8, M9, and M10) when all five rotating bifacial PV modules (M1, M2, M3, M4, and M5) are rotated to obtain the minimum transparency degree of the PV skylight (that is to say, the maximum shadowing of the underlying surface).

Finally, in order to make possible a comparative analysis with an "opaque" fixed PV skylight, Figure 13d plots the waveforms of the five short-circuit currents of the same five PV modules of the underlying PV surface (M6, M7, M8, M9, and M10) when all five rotating bifacial PV modules (M1, M2, M3, M4, and M5) are fixed at the 90° angle position.

The main outcomes of the analysis of Figure 13, from (a) to (d), can be summarized as follows.

The prototype specifically controlled for obtaining the maximum generation of electricity (Figure 13a) shows that the corresponding PV skylight also has a very low degree of transparency to the incident sunlight. Furthermore, the corresponding illuminance level on the underlying surface is quite uniform; only in the early morning and the late afternoon is the illuminance degree slightly higher on the first and the last lateral zones of the underlying surface because of the sunlight passing through from the bottom of the lateral sides of the PV skylight.

The prototype specifically controlled to obtain the maximum degree of transparency (Figure 13b) shows that it is possible to make a "fully transparent" prototype (like a conventional glass skylight with no PV generation) by specifically controlling its rotating PV modules. In fact, the curves of the short-circuit currents of the PV modules (M6 ÷ M10) of the underlying PV surface show that they can generate practically the same currents of the overlying rotating PV modules when these are fixed at the 90° angle position. Please note that this operating condition can be easily obtained by controlling the angular position of all the rotating PV modules so that they are constantly "parallel" to the incident sun rays. However, in this operating condition, the daily electricity generation of the PV skylight decreases to about 28% of the maximum generable value.

The prototype specifically controlled to obtain the minimum degree of transparency (that is to say, the maximum shadowing, Figure 13c) reveals that this specific operating condition of the PV skylight practically coincides with that of the maximum generation of electricity.

Finally, the prototype that emulates a fixed "opaque" PV skylight (Figure 13d) shows that this generates only 89% of the maximum generable electricity. Furthermore, its degree of transparency to the incident sunlight and the uniformity of the corresponding illuminance level on the underlying surface are practically the same as those of the rotating PV skylight generating the maximum electrical power.

## 6. Directions for Further Research and Conclusions

The main objective of this paper was to introduce and experimentally emulate the behavior of an innovative bifacial PV skylight endowed with an "under glass" fully protected single-axis solar tracker, whose basic idea was introduced with an Italian patent dated 2018 and a recent pending patent application dated 2023. The experimental investigation was developed by considering different operating conditions and the number of rotating PV modules installed side by side on the available surface area exposed to the sunlight to understand its optimal installation configuration before proceeding to the executive design and the construction of its definitive commercial version.

This PV skylight is intended to be profitably fully integrated into a building; therefore, its optimal installation configuration should be determined considering different performance parameters, including (i) the electricity generation capacity, (ii) the illuminance and shading capacity, and (iii) the controllability of the illuminance and shading. Furthermore, a proper analysis of the main costs (for its construction, operation, and maintenance) should also be conducted, together with an evaluation of the consequent payback time. Obviously, the cost analysis should consider the definitive constitutive version of the PV skylight

rather than simply considering this first "complex and flexible" prototype, which was homemade for scientific investigation purposes. For a useful and realistic evaluation of the payback time, the entire energetic benefits of the definitive PV skylight obtained during the years of its operation should be properly translated in terms of corresponding economic benefits. This strongly depends on the kind and the intended use of the building in which the PV skylight is installed. For the aforementioned reasons, the analysis of the building, operation, and maintenance costs of the proposed PV skylight together with its respective payback time is deferred to a future study that will address the executive design, realization, and utilization of the definitive version of the PV skylight, here, simply emulated using a prototype and preliminarily tested by experiments. In this way, the future study will make an overall view of the subject possible.

The homemade prototype offered the possibility to investigate two different installation configurations experimentally. The first is based on only three rotating PV modules installed side by side and visibly distanced from each other, and the second is based on five rotating PV modules installed side by side with no empty spaces between them, fully covering the entire available surface exposed to the sunlight.

By analyzing the experimental results of the first case study, it can be summarized that the prototype endowed with only three rotating bifacial PV modules, installed side by side and visibly distanced from each other:

(i)    Can generate almost +50% more electricity than a fixed semitransparent PV skylight based on three monofacial PV modules and almost +30% more electricity than a fixed semitransparent PV skylight based on three bifacial PV modules with the same transparency degree;

(ii)   Generates a maximum of electricity that is almost −11% lower than that generable by an "opaque" fixed monofacial PV skylight, occupying the same available surface with five fixed monofacial PV modules;

(iii)  Has a medium degree of transparency, which is always very good; nevertheless, it does not guarantee good uniformity of illuminance on the underlying surface. Furthermore, because of the presence of the two empty spaces between the three rotating PV modules, it is not capable of becoming sufficiently opaque to the incident sunlight to profitably control the illuminance level on the underlying surface.

By analyzing the experimental results of the second case study, it can be summarized that the prototype endowed with five rotating bifacial PV modules installed side by side without any empty space between them and practically occupying the whole available surface of the PV skylight:

(i)    Can generate about +11% more electricity than a fixed bifacial "opaque" PV skylight, about +22% more electricity than a fixed monofacial "opaque" PV skylight, and about +40% more electricity than a semitransparent PV skylight constructed with three rotating bifacial PV modules;

(ii)   Guarantees a high degree of controllability of its transparency, from the maximum value of a conventional transparent skylight (without any photovoltaic generation capacity) to the very low value of a conventional, almost opaque skylight;

(iii)  Guarantees a very good uniformity of the illuminance level on the underlying surface.

By comparing the results of the two different installation configurations and considering all the performance parameters that influence the effectiveness of the PV skylight, it is evident that the most attractive installation configuration is based on five rotating bifacial PV modules installed side by side with no empty spaces between them, occupying the entire available surface exposed to sunlight. One technical limitation of this configuration could be related to the previously cited reciprocal shadowing phenomena that affect the installed PV modules daily, especially during the early morning and the evening, because they are not spaced apart from each other. In principle, this phenomenon could cause adverse and repetitive hotspots on the shaded PV modules, which can significantly affect their lifetime. Nevertheless, without any prejudice to the opportunity of a further specific

study of the phenomenon, as explained in [31], in this specific context, there are at least two good reasons to believe that this phenomenon may not be harmful. First, the low power and the low number of series-connected PV cells of the involved PV modules should guarantee very low currents, very low reverse voltages, and, therefore, very low power dissipated on the shaded PV cells. Second, the shadowing always uniformly affects all the series-connected PV cells of all the single-row PV modules, and this means that, during an eventual hotspot, the whole dissipated power is equally distributed among all the PV cells shaded at the same time. Furthermore, since the shadowing is repetitive, predictable and known, in [31], it was shown that it can be simple to effectively cope with this phenomenon by taking advantage of the battery technology.

Finally, the latter configuration will be the starting point for future studies and for the executive design of the definitive commercial version of the proposed PV skylight. Additional studies will be addressed by the authors on the utilization of this kind of PV in the specific field of photovoltaic greenhouses.

## 7. Patents

This experimental study was realized by making explicit reference to an "industrial invention" whose details are fully specified within the documents of the Italian patent n. 0001430077, issued by the Italian Ministry of the "Sviluppo Economico" on 2 October 2018; the aforementioned patent is referred to in [28]. Also, some new additional ideas have been experimented with by taking advantage of the contents of a new pending Italian application patent N. 102023000011895, submitted by the authors on 9 June 2023; this additional application patent is referred to in [29].

**Author Contributions:** Conceptualization, R.C.; methodology, R.C.; software, C.B.; validation, C.B. and R.C.; formal analysis, R.C.; investigation, C.B. and R.C.; resources, C.B. and R.C.; data curation, C.B.; writing—original draft preparation, R.C.; supervision, R.C.; funding acquisition, R.C. All authors have read and agreed to the published version of the manuscript.

**Funding:** This work was funded by the Next Generation EU—Italian NRRP, Mission 4, Component 2, Investment 1.5, call for the creation and strengthening of 'Innovation Ecosystems', building 'Territorial R&D Leaders' (Directorial Decree n. 2021/3277)—project Tech4You—Technologies for climate change adaptation and quality of life improvement, n. ECS0000009. This work reflects only the authors' views and opinions; neither the Ministry for University and Research nor the European Commission can be considered responsible for them.

**Data Availability Statement:** The data presented in this study are available on request from the corresponding author.

**Conflicts of Interest:** The authors declare no conflict of interest.

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
