# Peer review of "A Building-Integrated Bifacial and Transparent PV Generator Operated by an “Under-Glass” Single Axis Solar Tracker"

_energies, doi:10.3390/en16176350_

Round 1
Reviewer 1 Report
The presented article describes a roof louver system based on double-sided photovoltaic modules for the simultaneous generation of electricity and shading of the interior space. The subject of the article is relevant and may be of interest to specialists and researchers in the fields of solar energy, solar tracking systems, autonomous power supply and sustainable development. The authors carried out some field test results, however, as remarks and recommendations, some points should be noted
1. All abbreviations in the text must be deciphered at the first mention of them.
2. Authors should add the "Methods and Materials" section to the content of the article.
3. Also, the authors should describe in more detail the scientific novelty of the proposed device, since many similar devices already exist. What are the advantages and disadvantages of the developed device compared to existing analogues?
4. It is not entirely clear the use of two elements instead of one double-sided, as stated in the title of the article, which worsens the economics of the project. The bottom element will have a minimum efficiency. How expedient is the use of EVA for bilateral operation of elements, when polysiloxane compounds improve both optical transparency and electrical efficiency, as well as the service life of photovoltaic converters (for example, 10.4018/IJEOE.2020040106 etc.) - in future studies, the authors should pay attention on sealed transparent components and high-efficiency double-sided photovoltaic converters.
5. What is the estimated cost of the developed system and its payback period?
6. In some places, the figures and the text describing them are difficult to understand - the authors should make simplifications, as well as consider the possibility of transferring some figures and descriptions to the appendices of the article.
7. In the article, little attention is paid to the tracking system for the position of the Sun - it should be described in more detail.
8. To enhance the scientific component of the work, the authors should have given theoretical calculations and formulas, the results of which should be compared with the obtained experimental values.
9. It is interesting to optimize the gap between the panels to ensure the minimization of shading of neighboring panels and improve the illumination of the interior space.
10. What practical recommendations do the authors give for the use of such a louvered system? Is it possible to standardize such a system somehow?
11. At the end of the work, the authors should indicate where and how they plan to use and implement the results.
12. Also, the authors should add a subsection "Directions for further research", where the authors should describe the planned work on the topic under consideration.
In general, the presented article leaves a positive impression, however, it is not without shortcomings. After eliminating these comments and taking into account the recommendations made, the presented article may be of interest to readers of the journal "Energies".
Authors should carefully proofread the text of the article to avoid spelling and punctuation errors, as well as seek help from a native English speaker for the final proofreading of the content of the text.
Author Response
Response to the comments of the Reviewer 1
First, the authors want to thank the Reviewer very much for his valuable comments, which are fully considered and used as a very helpful guideline for revising the paper, before its re-submission.
General reviewer’s comment: The presented article describes a roof louver system based on double-sided photovoltaic modules for the simultaneous generation of electricity and shading of the interior space. The subject of the article is relevant and may be of interest to specialists and researchers in the fields of solar energy, solar tracking systems, autonomous power supply and sustainable development. The authors carried out some field test results, however, as remarks and recommendations, some points should be noted.
Reviewer’s comment: 1. All abbreviations in the text must be deciphered at the first mention of them.
Author’s response: ok, done.
Reviewer’s comment: 2. Authors should add the "Methods and Materials" section to the content of the article.
Reviewer’s comment: 8. To enhance the scientific component of the work, the authors should have given theoretical calculations and formulas, the results of which should be compared with the obtained experimental values.
Author’s response: ok, done. According with comments and recommendations also of the Reviewer 4, in the revised paper we have included a new theoretical and designing addressed section 2., just named “Methods and Materials”.
Reviewer’s comment: 3. Also, the authors should describe in more detail the scientific novelty of the proposed device, since many similar devices already exist. What are the advantages and disadvantages of the developed device compared to existing analogues?
Author’s response: ok, done. Some more specific details are added to the original description of the proposed device and we hope that now its scientific novelty is clearer.
Reviewer’s comment: 4. It is not entirely clear the use of two elements instead of one double-sided, as stated in the title of the article, which worsens the economics of the project. The bottom element will have a minimum efficiency. How expedient is the use of EVA for bilateral operation of elements, when polysiloxane compounds improve both optical transparency and electrical efficiency, as well as the service life of photovoltaic converters (for example, 10.4018/IJEOE.2020040106 etc.) - in future studies, the authors should pay attention on sealed transparent components and high-efficiency double-sided photovoltaic converters.
Author’s response: As stated in the title of the paper, we fully agree that the bifaciality of the proposed PV-skylight has to be obtained by utilizing conventional/commercial bifacial PV-cells (i.e., based only on a single double-sided layer). Nevertheless, as mentioned in the paper, only for scientific investigation purposes, we decided to use two separate monofacial PV-strings for composing our self-made bifacial PV-strings. With some additional opportune details, this “unusual” solution was adopted just for having the possibility to experimentally analyze what is the specific contribution of the bottom face of a bifacial PV-string on the whole generation capability of a bifacial PV-skylight, under different operating conditions. In fact, by this way, we were able to measure the two currents separately generated, at the same time, by the top and the bottom faces of the self-made bifacial PV-strings. It is obvious that, for a future realization of a definitive “commercial” PV-skylight and outside the specific objectives of this scientific experimental study, the use of high-efficient and cheaper commercial bifacial PV-cells (e.g., bifacial PERC solar cells) remains a must. Also, for a future realization of a “commercial” PV-skylight, we will pay due attention to the possibility to profitably substitute the EVA with other encapsulating materials (e.g., the polysiloxane) for improving the optical and electrical performances of the bifacial PV-strings. Evidently, this relevant aspect was not clearly presented and explained in the original paper; thus, we have revised our writing, hoping now it clearer.
Reviewer’s comment: 5. What is the estimated cost of the developed system and its payback period?
Author’s response: To respond to a similar request of other reviewers, in the revised version of the paper, we have added some brief notes about the economics on the implementation of our proposed solution, within a new section 6., named "Directions for further research".
Reviewer’s comment: 6. In some places, the figures and the text describing them are difficult to understand - the authors should make simplifications, as well as consider the possibility of transferring some figures and descriptions to the appendices of the article.
Author’s response: ok, done. We revised the description of the figures, hoping that now they are clearer.
Reviewer’s comment: 7. In the article, little attention is paid to the tracking system for the position of the Sun - it should be described in more detail.
Author’s response: ok, done. We hope that now the description of the solar tracking is more suitable.
Reviewer’s comment: 9. It is interesting to optimize the gap between the panels to ensure the minimization of shading of neighboring panels and improve the illumination of the interior space.
Author’s response: In the revised version of the paper, this relevant aspect is analyzed and discussed in the new theoretical and designing addressed section 2., named “Methods and Materials”.
Reviewer’s comment: 10. What practical recommendations do the authors give for the use of such a louvered system? Is it possible to standardize such a system somehow?
Author’s response: As mentioned in the paper and depicted in the figure 3, in our opinion, one of the most profitable utilization, of our patent protected basic idea, is that of concretize it in the form a PV-skylight, to be mounted, preferably, on the rooftops of a variety of buildings. In that sense, it is realistic to imagine possible also its standardization, through a modular product solution which can be used as base for coping with different energetic and aesthetic needs.
Reviewer’s comment: 11. At the end of the work, the authors should indicate where and how they plan to use and implement the results.
Author’s response: ok, done. In the revised conclusion section, we have tried to better clarify also this practical aspect.
Reviewer’s comment: 12. Also, the authors should add a subsection "Directions for further research", where the authors should describe the planned work on the topic under consideration.
Author’s response: ok, done. In the revised paper, we have added a new section 6., just named "Directions for further research".

Reviewer 2 Report
The authors of the paper propose innovative solutions that can radically change the architecture of modern buildings. A new type of photovoltaic generator is fully integrated into the building. It not only generates electricity, but also helps to improve both the overall energy performance and the aesthetic appearance of the building.
The authors have investigated the various variants of the proposed technical solution in great detail and presented the results of the investigations carried out.
The article is of interest to a wide range of specialists in electrical engineering and architecture.
I recommend the article for publication.
However, there remain questions of economics on the implementation of the proposed technical solutions.
Author Response
Response to the comments of the Reviewer 2
First, the authors want to thank the Reviewer very much for his valuable comments, which are fully considered and used as a very helpful guideline for revising the paper, before its re-submission.
General reviewer’s comment: The article is of interest to a wide range of specialists in electrical engineering and architecture. I recommend the article for publication. However, there remain questions of economics on the implementation of the proposed technical solutions.
Author’s response: Thanks again for your kind comments. To respond to a similar request of other reviewers, in the revised version of the paper, we have added some brief notes about the economics on the implementation of our proposed solution, within a new section 6., named "Directions for further research".

Reviewer 3 Report
This paper has introduced a good topic related to renewable energy. However, it requires extensive editing before it becomes ready for publication.
1. The abstract is long and needs to be focused on the overall work.
2. The number of references in the introduction is low and needs more recent work to be added.
3. Please define PV in your abstract. You need to define it as Photovoltaic (PV).
4. please use formal language, in line 51 That said, this paper introduces, you can say " This paper introduces".
5. Figure 1 caption is very long.
6. Figure 8 caption is long as well.
7. How did you check your result? You need to compare the obtained result with previous work.
8. Result section is very wordy.
9. Do not use We, Our, Us.
10. Please state your work limitation in the conclusion.
Extensive editing of English language required
Author Response
Response to the comments of the Reviewer 3
First, the authors want to thank the Reviewer very much for his valuable comments, which are fully considered and used as a very helpful guideline for revising the paper, before its re-submission.
Reviewer’s comment: 1. The abstract is long and needs to be focused on the overall work.
Author’s response: ok, done. We hope that now the abstract is better.
Reviewer’s comment: 2. The number of references in the introduction is low and needs more recent work to be added.
Author’s response: ok, done. We have considered and added eleven new recent references.
Reviewer’s comment: 3. Please define PV in your abstract. You need to define it as Photovoltaic (PV).
Author’s response: ok, done.
Reviewer’s comment: 4. please use formal language, in line 51 That said, this paper introduces, you can say " This paper introduces".
Reviewer’s comment: 9. Do not use We, Our, Us.
Author’s response: ok, done. We hope that now the language of the revised paper is more formal.
Reviewer’s comment: 5. Figure 1 caption is very long.
Reviewer’s comment: 6. Figure 8 caption is long as well.
Author’s response: ok, done. We hope that now the captions of the figures 1 and 8 are ok.
Reviewer’s comment: 7. How did you check your result? You need to compare the obtained result with previous work.
Author’s response: ok done. In the revised writing of the paper, we have inserted some more express comparisons with some other installation configurations similar to our patent protected basic idea.
Reviewer’s comment: 8. Result section is very wordy.
Author’s response: ok, done. We hope that now the result section is more suitable.
Reviewer’s comment: 10. Please state your work limitation in the conclusion.
Author’s response: ok, done. We hope that now the conclusion section is more suitable.

Reviewer 4 Report
In this paper, the authors proposed an innovative rotating and bifacial PV-modules which are specifically made to be installed “under-glass”, within a likewise custom-made transparent casing. A low-power home-made prototype is tested. The structure and design of the proposed BIPV-generator are interesting. However, following comments are suggested.
1-As shown in Fig. 1, the home-made bifacial PV-module prototype is composed of two independent PV-string. Why do not the authors directly use the bifacial solar cells to form a PV-string, e.g., bifacial PERC solar cells? Thus, the middle layer EVA is no longer needed, and total number of layers of the home-made bifacial PV-module can be reduced.
2-For the designed bifacial PV-module prototype, the material of front sheet and back sheet is plastic, instead of commonly used glasses. However, the detail reflectance of the applied plastic front sheet and plastic back sheet is not presented. Does the spectral response of plastic match that of solar cells? The mismatched spectral response may cause low efficiency of the PV module.
3-Two cases are experimentally studied. However, more theoretical analysis is required. For instance, how to determine the suitable width of empty spaces between adjacent rotating axes? How much is the ratio of reflection irradiance between different rotating axes of bifacial PV modules? What is the function relationship between I-V characteristic of designed bifacial PV module with ambient condition, e.g., tilt angle, irradiance and temperature? How to theoretically estimate the power of designed bifacial PV module? The theoretical analysis would help readers to know how the rotating bifacial PV modules are designed and develop more suitable improvements of the BIPV-generator. And the most important, is the proposed BIPV-generator more cost-effective than other forms of BIPV?
4-The length of this paper must be shrunk to highlight the novelty of the proposed idea.
Author Response
Response to the comments of the Reviewer 4
First, the authors want to thank the Reviewer very much for his valuable comments, which are fully considered and used as a very helpful guideline for revising the paper, before its re-submission.
Reviewer’s comment: 1. As shown in Fig. 1, the home-made bifacial PV-module prototype is composed of two independent PV-string. Why do not the authors directly use the bifacial solar cells to form a PV-string, e.g., bifacial PERC solar cells? Thus, the middle layer EVA is no longer needed, and total number of layers of the home-made bifacial PV-module can be reduced.
Author’s response: As mentioned in the paper, we decided to use two independent monofacial PV-strings for composing our bifacial PV-string protypes exclusively for having the possibility to experimentally and separately analyze the contribution of the “rear face” of the bifacial PV-strings on the whole generation capability of the proposed PV-skylight, under different operating conditions. We also underlined that, for a future realization of a “commercial” PV-skylight, we strongly recommend to use exclusively high-efficient and cheaper commercial bifacial PV-cells (e.g., bifacial PERC solar cells). Of course, in the future “commercial” PV-skylight, the middle layer EVA is no longer needed and the total number of layers could be profitably reduced. Evidently, this relevant aspect was not clearly presented and explained in the original paper; thus, we have revised our writing, hoping now it clearer.
Reviewer’s comment: 2. For the designed bifacial PV-module prototype, the material of front sheet and back sheet is plastic, instead of commonly used glasses. However, the detail reflectance of the applied plastic front sheet and plastic back sheet is not presented. Does the spectral response of plastic match that of solar cells? The mismatched spectral response may cause low efficiency of the PV module.
Author’s response: Yes, for the realization of the front sheet and of the back sheet of our Bifacial PV-strings we used a transparent plastic film. With some more details, we used (and recommend to use) specific high-performance PET laminates characterized by ultra-high transparency, which are already commercially known for being a valid alternative to the glass, in particular, for the realization of low-thickness and light bifacial PV-modules. They also feature an innovative and reliable coating which protects the module from scratches, abrasion, corrosion and UV thus enhancing the module durability and performance over time; it is also an anti-glare coating, helping to catch and diffuse more light, especially during sunrise and sunset when the sun is not at zenith position. We decided to not explicitly cite the commercial plastic material we used for our prototype just to avoid inappropriate advertising issues. Nevertheless, we added some technical details in the revised paper, while, for your convenience, the complete datasheet of the used material can be found at the following link: https://www.coveme.com/files/documenti/2022-brochure/solare/leaflet_4_ante_solare_2023_web-bis.pdf.
Reviewer’s comment: 3. Two cases are experimentally studied. However, more theoretical analysis is required. For instance, how to determine the suitable width of empty spaces between adjacent rotating axes? How much is the ratio of reflection irradiance between different rotating axes of bifacial PV modules? What is the function relationship between I-V characteristic of designed bifacial PV module with ambient condition, e.g., tilt angle, irradiance and temperature? How to theoretically estimate the power of designed bifacial PV module? The theoretical analysis would help readers to know how the rotating bifacial PV modules are designed and develop more suitable improvements of the BIPV-generator. And the most important, is the proposed BIPV-generator more cost-effective than other forms of BIPV?
Author’s response: ok, done. According with comments and recommendations also of the Reviewer 1, in the revised paper we have included a new theoretical and designing addressed section 2., named “Methods and Materials”. Some additional details, about the economics on the implementation of our proposed solution, are also included in a new section 6., named "Directions for further research".
Reviewer’s comment: 4. The length of this paper must be shrunk to highlight the novelty of the proposed idea.
Author’s response: Considering the specific request to add a new theoretical/designing section and also additional requests of other Reviewers (i.e. about the improvement of the references), it was hard trying to shrunk more the revised paper.

Round 2
Reviewer 1 Report
The authors have done some work to supplement and correct the article. Before the final upload of the article, the authors should carefully proofread the text, eliminate spelling and punctuation errors, shorten and simplify the wording.
Proofreading by a native English speaker is required.
Reviewer 3 Report
The Authors answered all my questions
Overall is better than the first draft
Reviewer 4 Report
The authors have revised accordingly.